# Chromosome-level genome assembly and population genomic resource to accelerate orphan crop lablab breeding

Isaac Njaci [1,2,10], Bernice Waweru [1,10], Nadia Kamal [3,10], Meki Shehabu Muktar [4,10], David Fisher [5], Heidrun Gundlach [3], Collins Muli[1], Lucy Muthui[1], Mary Maranga[6], Davies Kiambi[7], Brigitte L. Maass [8], Peter M. F. Emmrich[2,9], Jean-Baka Domelevo Entfellner [1], Manuel Spannagl [3], Mark A. Chapman [5] ✉, Oluwaseyi Shorinola [1,2] ✉ & Chris S. Jones [1] ✉

Under-utilised orphan crops hold the key to diversified and climate-resilient food systems. Here, we report on orphan crop genomics using the case of *Lablab purpureus* (L.) Sweet (lablab) - a legume native to Africa and cultivated throughout the tropics for food and forage. Our Africa-led plant genome collaboration produces a high-quality chromosome-scale assembly of the lablab genome. Our assembly highlights the genome organisation of the trypsin inhibitor genes - an important anti-nutritional factor in lablab. We also re-sequence cultivated and wild lablab accessions from Africa confirming two domestication events. Finally, we examine the genetic and phenotypic diversity in a comprehensive lablab germplasm collection and identify genomic loci underlying variation of important agronomic traits in lablab. The genomic data generated here provide a valuable resource for lablab improvement. Our inclusive collaborative approach also presents an example that can be explored by other researchers sequencing indigenous crops, particularly from low and middle-income countries (LMIC).

Three major crops currently provide more than 40% of global calorie intake[1]. This over-dependence on a few staple crops increases the vulnerability of global food systems to environmental and social instabilities[2]. One promising strategy to diversify food systems is to improve the productivity and adoption of climate-resilient but underutilised orphan crops through genome-assisted breeding[3].

*Lablab purpureus* L. Sweet, (hereafter referred to as lablab, Fig. 1a) is an indigenous African legume that is remarkably drought-resilient and thrives in diverse environments. As such it is widely cultivated throughout the tropical and subtropical regions of Africa and Asia[4]. Lablab is a versatile multipurpose crop that contributes towards food, feed, nutritional and economic security, and is also rich in bioactive compounds with pharmacological potential, including against SARS-Cov2[5–8]. Climate change is driving researchers to investigate crops like lablab for their outstanding drought tolerance[9].

Genome-assisted breeding offers hope of a new green revolution by helping to uncover and unlock novel genetic variation for crop improvement. Over the last 20 years, the genomes of 135 domesticated

[1]International Livestock Research Institute, PO Box 30709-00100 Nairobi, Kenya. [2]John Innes Centre, Norwich Research Park, Norwich NR4 7UH, UK. [3]Helmholtz Zentrum München, Plant Genome and Systems Biology, Ingolstädter Landstr. 1, 85764 Neuherberg, Germany. [4]International Livestock Research Institute, Addis Ababa, Ethiopia. [5]University of Southampton, School of Biological Sciences, Southampton SO17 1BJ, UK. [6]Department of Biochemistry, Jomo Kenyatta University of Agriculture and Technology, Nairobi 00200, Kenya. [7]Bioscience Research Centre (PUBReC), Pwani University, P.O Box 195-80108 Kilifi, Kenya. [8]Department of Crop Sciences, Georg-August-University Göttingen, Grisebachstr 6, 37077 Göttingen, Germany. [9]Department for International Development, University of East Anglia, Norwich NR4 7TJ, UK. [10]These authors contributed equally: Isaac Njaci, Bernice Waweru, Nadia Kamal, Meki Shehabu Muktar. ✉e-mail: m.chapman@soton.ac.uk; shorinolao@gmail.com; c.jones@cgiar.org

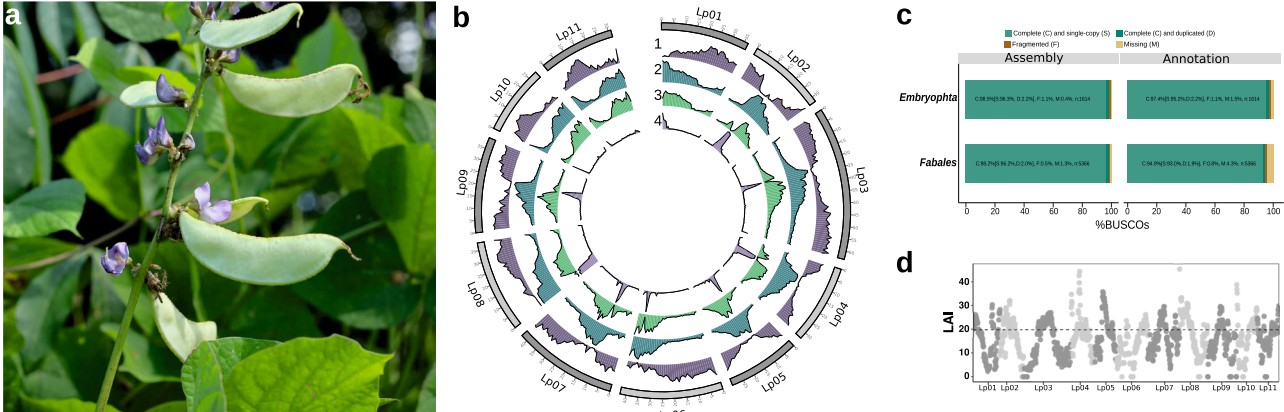

**Fig. 1 | Genome assembly of lablab. a** *Lablab purpureus* plant showing flowers, leaves and pods. **b** Gene and repeat landscape of the lablab genome. The tracks from the outer to the inner track show 1) gene density, 2) repeat density, 3) LTR-RT density, 4) tandem repeat density. **c** BUSCO scores of the lablab genome and gene annotation using the embryophyta and fabales reference lineages. **d** LAI index of the 11 lablab chromosomes. Source data are provided as a Source Data file.

crops have been sequenced and assembled[10], including those of orphan crops[3]. A draft genome for lablab was previously produced using a short read sequencing approach[11]. However, it has recently been acknowledged that researchers from Africa are grossly under-represented in the genome sequencing efforts of their indigenous orphan crops[10,12]. This has primarily been due to the acute lack of sequencing facilities and high-performance computing infrastructures as well as bioinformatics capacity to handle big genome data[13].

Here, we present an African-led genome collaboration that over-comes this under-representation through an inclusive orphan crop genomics approach. Our Africa-led genome collaboration produces a long read-based chromosome-scale assembly of lablab. We confirm the dual domestication origin of lablab by resequencing wild and domesticated accessions. We also examine genetic and phenotypic diversity in a comprehensive lablab collection. Finally, we discuss the main features and benefits from our inclusive orphan crop genomics approach and suggest that this can serve as a roadmap for future genomic investigations of indigenous African crops.

## Results
### Genome sequencing
High acquisition and maintenance cost of sequencing platforms is a major limiting factor to genomics research in Africa. To circumvent this limitation, we used the portable and low-cost Oxford Nanopore Technology (ONT) MinION platform for in-country sequencing of the genome of lablab (cv. Highworth). We generated 4.7 M reads with a mean read length of 6.1 kb (Supplementary Table 1). This amounted to 28.4 Gbp of sequences and 67x coverage of the lablab genome based on a previously estimated genome size of 423 Mbp[11]. The reads were initially assembled into 2260 contigs with an N50 of 11.0 Mbp and total assembly length of 426.2 Mbp. The assembly was polished for error correction using ~380x of publicly available Illumina short reads pre-viously generated from the same cultivar.

Using High-throughput Chromosome Conformation Capture (Hi-C)[14], we scaffolded the contigs into 11 pseudomolecules repre-senting the 11 lablab chromosomes (Fig. 1b, Supplementary Fig. 1) and covering 417.9 Mbp (98.03% of total assembled sequence and 98.6% of the estimated genome size), with an N50 of 38.1 Mbp (Supplementary Table 2) and BUSCO completeness scores of 98.5% and 98.2% against the embryophyta and fabales lineages, respectively (Fig. 1c). For con-sistency with published legume genome sequences, we assigned chromosome names to the Hi-C-scaffolded pseudomolecules based on syntenic relationship with closely related legumes—*Phaseolus vulgaris*

(common bean[15]) and *Vigna unguiculata* (cowpea[16]) (Supplemen-tary Fig. 2).

### Genome annotation and gene family analyses
We used an automated pipeline based on protein homology, transcript evidence and ab initio predictions to identify protein-coding genes in the lablab genome. This resulted in a total of 30,922 gene models. On average each gene had 2.57 isoforms resulting in a total of 79,512 transcripts. A subset of 24,972 of these gene models show no homol-ogy to transposable elements (TEs) and can be confidently considered as high-quality protein-coding non-TE gene models (Fig. 1b, Supple-mentary Table 3). BUSCO scores of the TE-filtered gene models showed that 97.4%, and 94.9% of the universal single copy genes from the embryophyta and fabales, respectively, were complete, suggesting a high level of completeness of the gene space (Fig. 1c). We found expression support for 73% of these gene models using RNASeq data from four tissues[11]. Functional descriptions could be assigned to 28,927 (93.3%) of the genes. In addition to the protein-encoding gene models, we detected 542 tRNA-encoding genes.

A total of 168,174 TE sequences, occupying 28.1% of the genome, were identified in the lablab genome (Fig. 1b). Of these, 89.6% were classified into 13 superfamilies and 2353 known families (Supplemen-tary Table 4, Supplementary Fig. 3). Long Terminal Repeat-RetroTransposons (LTR-RTs) were the most abundant TEs, with 85,149 sequences occupying 83 Mb (19.9%) of the genome (Fig. 1b). Copia was the most abundant LTR-RT superfamily, occupying 13.2% of the genome compared to gypsy elements, which occupied only 4.7%. We also report an average LTR Assembly Index (LAI) of 19.8 (Fig. 1d) which, according to the classification system proposed by ref. [17], indicates a reference quality assembly with a high level of contiguity of the repetitive and intergenic regions. DNA transposons were smaller in number and size relative to LTR-RTs, and were distributed more evenly across the chromosomes (Supplementary Table 4, Supplemen-tary Fig. 3).

A further 100,741 repetitive sequences were identified but could not be classified as TEs. Combining the annotated TEs and unclassified repeats reveals an overall repeat content of 43.4% of the genome. We also identified 142,302 tandem repeats (TRs, Fig. 1b, Supplementary Table 5). Most of these were minisatellites (10–99 bp), while satellite repeats (>100 bp) make up the largest total proportion of TRs in the genome (Supplementary Table 5). Both the tandem and unclassified repeats were found to concentrate within a distinct, overlapping cluster at the point of peak repeat density on each chromosome,

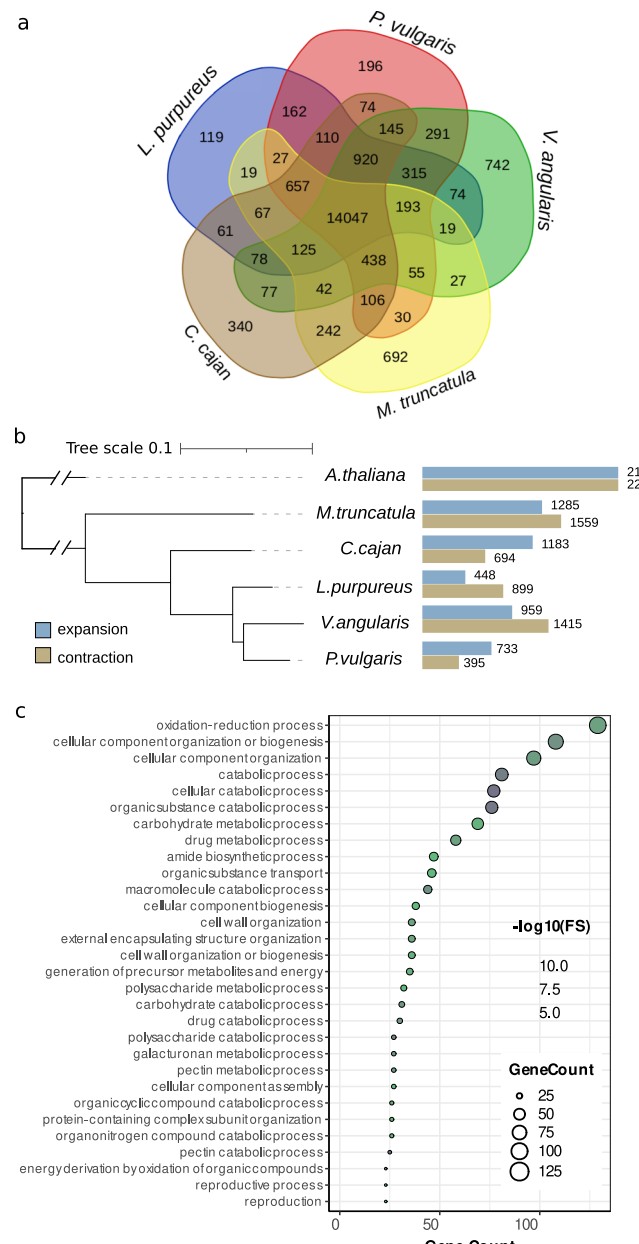

**Fig. 2 | Gene families in lablab. a** Venn diagram of the number of gene families common among and unique to lablab, *Phaseolus vulgaris, Vigna angularis, Medicago truncatula*, and *Cajanus cajan*. **b** Cladogram of the analysed species showing the number of expanded and contracted gene families in each. Figure constructed with iTol[93]. **c** Gene ontology terms enriched in the set of expanded gene families in *Lablab purpureus*. Source data are provided as a Source Data file.

indicating that they are likely centromeric repeats (Supplementary Fig. 3a).

Gene family analysis and comparison to other legumes (*P. vulgaris, V. angularis, Cajanus cajan, Medicago truncatula*), and using *Arabidopsis thaliana* as an outgroup, placed 24,397 (97.7%) of the 24,972 TE-filtered lablab genes into orthogroups (Supplementary Data 1). Comparison of the five legumes (Fig. 2a) revealed 14,047 orthogroups in common, and identified 417 (1.7%) lablab genes in 119 species-specific orthogroups that were absent from the other four legumes. These lablab-unique gene families were enriched for fatty acid biosynthesis, arabinose metabolism gene ontology (GO) classifiers, while several were involved in pollen–pistil interactions and general plant development (Supplementary Data 2). Using the

phylogenetic relationships between the species, 448 gene families were significantly expanded in lablab compared to other legumes and *Arabidopsis*, while 899 were contracted (Fig. 2b). Expanded gene families were enriched for lignin and pectin metabolism, and photosynthesis among others (Supplementary Data 3; Fig. 2c), while contracted gene families were involved in e.g. amide biosynthetic and metabolic processes (Supplementary Fig. 4).

Legume trypsin inhibitors are protease inhibitors that play important defence roles against pathogen and herbivorous insects[18], but are also considered as antinutritional factors that reduce digestibility and nutrient availability in legumes like lablab used as feed or fodder[19]. We catalogued the trypsin inhibitor (TI) gene family in lablab and identified 35 genes (Fig. 3), which represents enrichment for TI genes in the lablab genome compared to *A. thaliana* and legumes such as *M. truncatula* and *C. cajan* (Fig. 3a and b). Most of the genes are located in clusters on chromosomes Lp01, Lp04, Lp06, and Lp11, with two major clusters on chromosome Lp04 containing 21 (60%) trypsin inhibitor genes (Fig. 3c). A total of 23 of the 35 trypsin inhibitor encoding genes (66%) identified in the lablab genome are members of tandemly duplicated gene arrays. The large TI cluster on Lp04 shows synteny with *V. angularis, M. truncatula, C. cajan*, and *P. vulgaris*, while the five-gene cluster on Lp06 shows synteny to only *V. angularis* (Supplementary Fig. 5). We conclude that the trypsin inhibitor gene family is largely syntenous across legumes, with additional clusters and gene duplications specific to Lablab and *V. angularis*. Members of the Lp01, Lp06, and Lp011 gene clusters are highly expressed in vegetative tissues (leaves and stem), while 17 of the 21 trypsin inhibitor genes in the Lp04 cluster show a relatively low level of expression across both vegetative and reproductive tissues (Fig. 3d). These results highlight the usefulness of our assembly in dissecting the genomic architecture of genes underlying important traits for lablab improvement.

## Dual origin of domesticated lablab

Understanding the transition from wild species to domesticated crop can provide insight into the location of domestication, the strength of genetic bottleneck (and identification of wild alleles not present in the domesticated gene pool) and can lead to identifying candidate genes underlying domestication traits. Previous work has suggested that lablab domestication occurred at least twice, separately in the two-seeded and four-seeded gene pools[20,21]. Using our chromosome-scale assembly as a reference, we examined whether this is indeed the case by resequencing a panel of two-seeded and four-seeded wild (ssp. *uncinatus*) and domesticated (ssp. *purpureus*) lablab accessions (Supplementary Data 4). We also sequenced two individuals of subsp. *bengalensis* with 6-8 seeds per pod, two wild-like (likely feral) samples from India and one individual of *Dipogon lignosus* (L.) Verdc. (used as an outgroup). In addition, we gathered publicly available short-read data for cv. Highworth. All lablab samples had a >94% mapping against the lablab reference genome at a depth of 6.19–15.85x, while the outgroup had lower mapping (50.39% at a depth of 2.75x; Supplementary Data 4). The average coverage of lablab samples was 84.93% and for the outgroup 38.80% (using *-Q 13* and *-q 10*; Supplementary Data 4). Mapping and coverage was notably lower for the two-seeded accessions than the four-seeded accessions suggesting genomic divergence between these two gene pools. A total of 25,940,192 variants (23.7 M SNPs and 2.2 M indels) were identified across all 23 samples. After removing sites with missing data in more than two samples, this resulted in 9,797,710 variants (9.0 M SNPs and 0.8 M indels).

A filtered data set of 157,913 variants (>2 kb apart; see Methods) was used for phylogenetic analysis. Neighbor Joining phylogenetic analysis rooted with *Dipogon* revealed a clear division between the two- and four-seeded lablab samples (100% bootstrap support) with wild and domesticated samples found in both groups (Fig. 4). A parallel analysis using only variants from genes which had orthologues in *V. angularis, M. truncatula, C. cajan*, and *P. vulgaris* gave the same

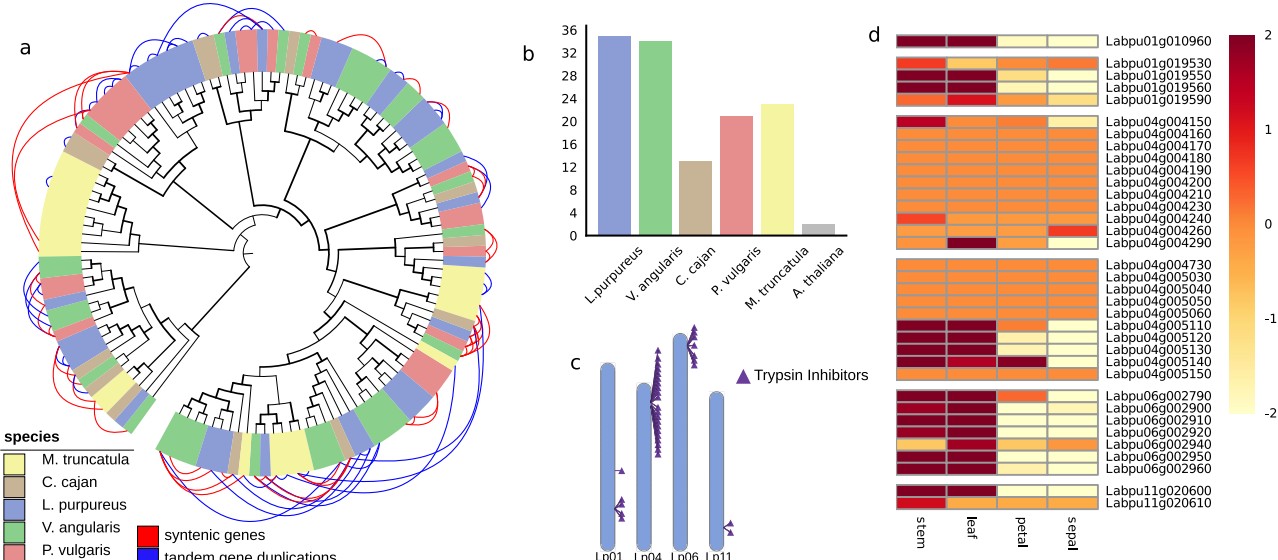

**Fig. 3 | Trypsin inhibitor gene family in lablab. a** Phylogeny of the trypsin inhibitor encoding genes in five different legume species including lablab; branch thickness corresponds to bootstrap values and increases with higher bootstrap; tree is rooted with the most divergent sequence from Arabidopsis; outer blue connections: tandemly duplicated genes, outer red connection: syntenic collinear genes. **b** Copy number of trypsin inhibitor genes in different plant species. **c** Organisation of the trypsin inhibitor gene clusters in the lablab genome. **d** Expression of the trypsin inhibitor genes in four different tissues. Genes in clusters are grouped in the heatmap. The variance stabilising transformed (vst) TPM levels correlate with the intensity of yellow to red colouring. Source data are provided as a Source Data file.

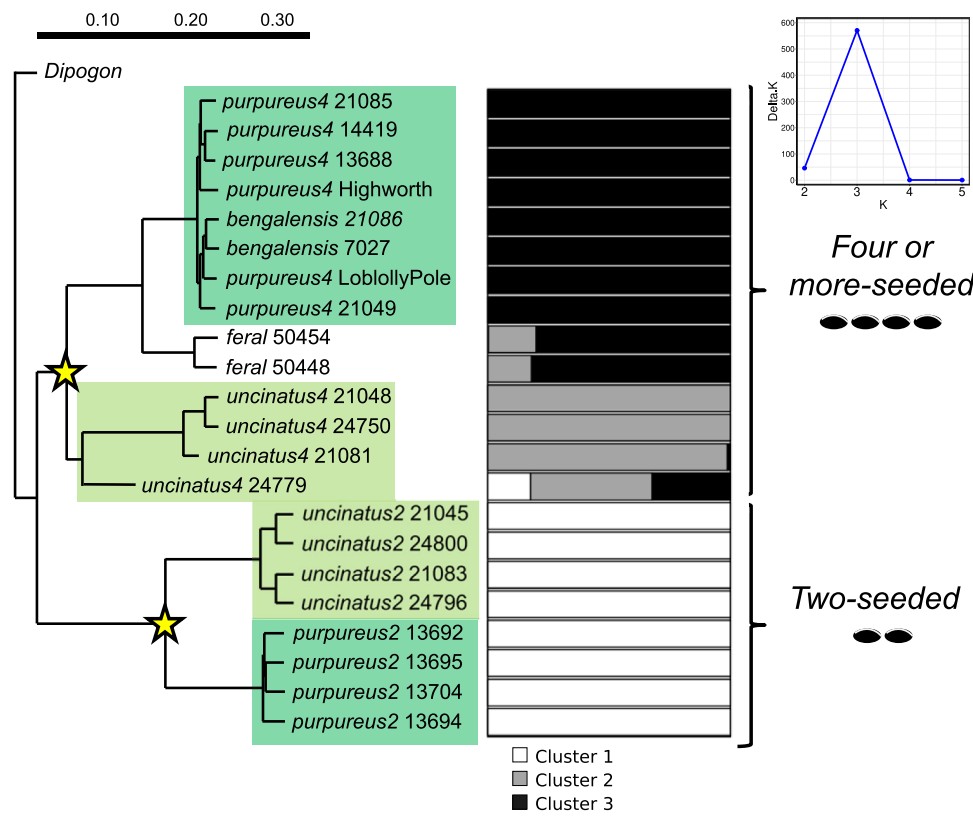

**Fig. 4 | Phylogenetics of lablab. Left**, Neighbor Joining phylogenetic relationships among lablab samples (2-seeded and 4-seeded *purpureus* (domesticated) and *uncinatus* (wild) subspecies) rooted on *Dipogon lignosus*. All nodes received full (100%) bootstrap support. Asterisks indicate the two domestication events. **Right**, STRUCTURE analysis of the same samples. The optimum number of clusters (K) was determined to be three (upper right), which are indicated as white, grey and black bars. Source data are provided as a Source Data file.

topology. Subsp. *bengalensis*, with 6-8 seeds per pod, is resolved as derived from the four-seeded group, and the likely feral accessions from India with a wild-like morphology are related to the four-seeded domesticates. We also carried out STRUCTURE analysis to examine the potential number of underlying genetic populations in the samples, which suggested $K = 3$ clusters best fit the data (Fig. 4). The two-seeded and four-seeded groups were again clearly differentiated, with no evidence of admixture between them. The four-seeded accessions were further differentiated into wild and cultivated groups. The likely feral accessions appear to be introgressed with wild (four-seeded) alleles. Other values of $K$ (the number of clusters) differentiate the feral accessions from the other four-seeded accessions (Supplementary Fig. 6). Our study thus confirms, based on multiple lines of evidence, the previous hypothesis of two origins of domesticated lablab.

The total number of variants for the four-seeded and two-seeded gene pools (and excluding the outgroup) was 10,666,655 and 5,200,923 variants, respectively. Genetic diversity ($\pi$ per 100 kb window based on only variant sites) was significantly greater (two-sided unpaired $t$-test, $t = 30.43$, df $= 8095$, $P < 0.001$) in the four-seeded group (0.00790 ± 0.00311 [SD]) than the two-seeded group (0.00599 ± 0.00260 [SD]). Divergence between the two- and four-seeded gene pools was high (mean $F_{ST}$ per 100 kb window = 0.438 ± 0.059 [SD]), which could suggest that these gene pools should be taxonomically reassessed as distinct taxa.

## Genetic diversity and association in a comprehensive lablab collection

We used a collection of 203 lablab accessions acquired since 1982 at the ILRI forage genebank to access diversity in the lablab gene pool. The collection comprises accessions acquired from 25 countries in Africa, Asia, Australia, Americas and Europe, with a substantial portion from Africa, which is believed to contain the highest genetic diversity and is the centre of diversity[21] (Ethiopia). Given the historic nature of this collection, we took a stepwise approach to characterise the accessions. First, we examined genetic purity within the accessions by genotyping 2300 individual plants across the 203 accessions using DArTseq genotyping-by-sequencing (GBS). This identified 42,494 genome-wide SNP and 36,803 SilicoDArT markers (described in Methods), of which 92% and 83% mapped onto the lablab genome, respectively (Supplementary Fig. 7). Based on Identity by Descent analysis using individual genotype data, we retained 191 accessions that were considered true-to-type (see Supplementary Note 1). Mean Nei's D between individuals of the 46 accessions with ≥2 individuals per accession ranged from 0.0015 to 0.1152 (median = 0.0396) indicating that within accession genetic diversity is generally low, as expected for a predominantly self-pollinating species such as lablab. Where multiple true-to-type individuals were found in an accession, the individual with the least missing data was selected for diversity analysis.

Using a subset of 7780 quality-filtered genome-wide SNPs (see Methods), we analysed population structure and clustering using STRUCTURE[22], which clearly suggested two population clusters in the lablab germplasm collection (Fig. 5a, Supplementary Fig. 8) supporting the two gene pools identified above with the whole genome resequencing. Further analysis of other values of K with some support (K = 3, 5, and 7) shows separation of the 4-seeded germplasm into smaller clusters. Based on K = 7, which separated the 4-seeded gene pool into groups that roughly correspond with geographic and morphological variation, similar clustering and population stratification were detected by hierarchical clustering and PCA (Fig. 5b and c, Supplementary Data 5). Clusters I and V contained accessions from Africa only, comprising all wild four-seeded and two-seeded accessions, respectively. Accessions in cluster III and VII were mainly from Asia (largely India) and Australia, and Cluster III included most of the accessions of ssp. *bengalensis*, which has long, relatively narrow pods

with up to eight seeds and a particular seed arrangement in the pod. Clusters II, IV and VI are from diverse origins and were mostly acquired by the Grassland Research Station (GRS, Kitale, Kenya). Pairwise $F_{ST}$ among the seven clusters varied from 0.31 between clusters III and IV to 0.97 between clusters I and II (Supplementary Table 6). Analysis of molecular variance (AMOVA) further showed the presence of greater genetic variation between (81%) than within clusters (19%) (Supplementary Table 7). The mean within-cluster genetic distance between accessions, Nei's D[23], was lowest for cluster II (mean $D = 0.002$) and highest for cluster I (mean $D = 0.186$; Supplementary Table 8). The low Nei's D in clusters VII (mean $D = 0.009$) and II probably reflects that they both contain several potential duplicate accessions of commercial provenance and the high Nei's D in cluster I likely reflects the group comprising both wild and domesticated accessions from Ethiopia.

The population clusters differed in several phenotypes based on historical phenology and morpho-agronomic trait data[24]. Cluster V, which comprised only the morphologically distinct wild (ssp. *uncinatus*) four-seeded accessions, was excluded due to lack of phenotype data. The 14 quantitative traits (Supplementary Fig. 9; Supplementary Table 9) and six out of seven qualitative characters (Supplementary Fig. 10; Supplementary Table 10) differed significantly among the six remaining genetic clusters of predominantly domesticated accessions, despite a certain level of phenotypic variation within every cluster. Cluster I includes accessions with short and wide pods containing two very large seeds, except for the wild two-seeded accessions (ssp. *uncinatus*) with small seed size. The plants in this group were late maturing and produced few but only coloured flowers, with shorter peduncles and longer flower rachis. Cluster II comprises the most homogeneous group; it had the tallest accessions that were late flowering with mostly white flowers and were broad and leafy with long flower peduncles, a high number of flowering nodes and four relatively small tan-coloured seeds per pod. Cluster III accessions, which included most of ssp. *bengalensis*, are phenotypically variable, containing early-flowering, short and mostly decumbent plants with four to six relatively large seeds per pod. Clusters IV and VI include diverse phenotypes; overall plants were broad, leafy and had intermediate plant height and flowering time and with larger leaves and shorter pods with up to four rather small seeds.

We combined the aforementioned historic phenotype data for 125 lablab accessions with quality-filtered DArTseq markers (7780 SNPs and 14,202 SilicoDArTs) for a genome-wide association study (GWAS). We identified 18 markers (8 SNPs and 10 SilicoDArTs) across eight chromosomes (Fig. 6, Supplementary Fig. 11) that are significantly associated with leaf length, leaf width, leaf ratio, plant height, days to 50% flowering, pod length, pod width, pod ratio, and thousand seed weight traits (Supplementary Table 11). The associated markers explained 7–24% of phenotypic variation (Supplementary Table 11). The identified markers will be useful in the genetic improvement of lablab through the application of marker-assisted selection and for further characterisation and map-based cloning of the QTLs.

## Discussion

Africa has rich plant biodiversity that includes 45,000 species[25], most of which are under-studied and many are not fulfilling their full potential. To fully explore these genetic resources, it is important to develop inclusive research models that enable and empower local researchers to study these species under a resource-limited research setting. Our work describes an inclusive African-led effort to produce a high-quality reference genome for a climate-resilient and multi-purpose native African orphan crop - lablab. Our chromosome-scale assembly of lablab improves on the previous assembly in several ways and allows us to highlight some interesting and important features of the lablab genome, and its domestication and population diversity.

With the use of long-reads and Hi-C scaffolding, we achieved a 61-fold improvement in contiguity and identified a further 34 Mbp of

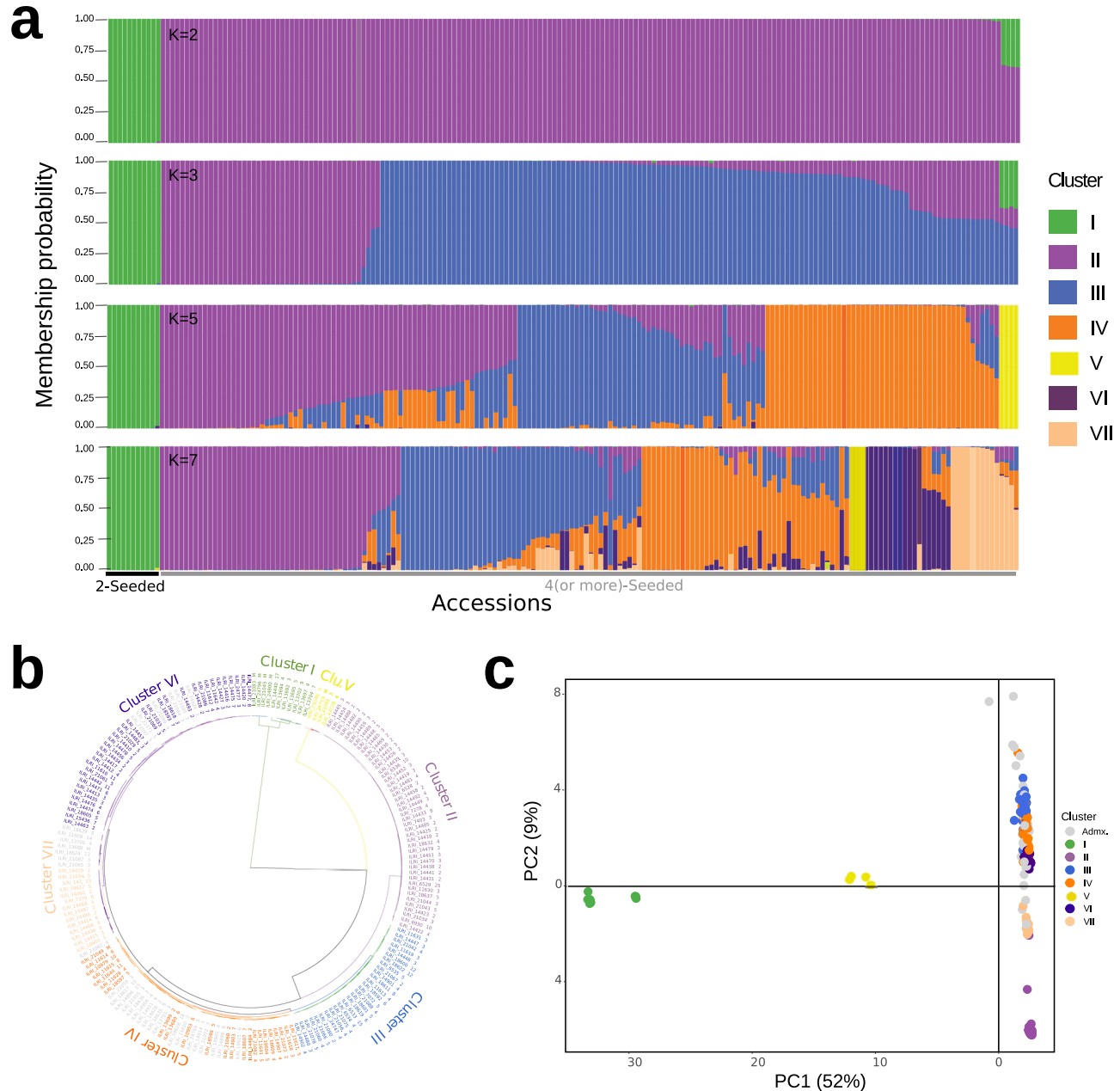

**Fig. 5 | Population genetic analysis of lablab. a** Bar plots based on the admixture model in STRUCTURE for multiple *K* (Membership of individual accessions to each subgroup is given in Supplementary Table 6). **b** Clusters detected by hierarchical clustering. **c** Clusters detected by PCA. The colours in (**b**) and (**c**) are according to the STRUCTURE *K* = 7 in (**a**). Accessions in admixture groups shown by light grey colour in (**b**) and (**c**). Source data are provided as a Source Data file.

repetitive sequences compared to the short-read based assembly[11]. In addition, the high average LAI[17] (19.8; Fig. 1d), comparable to the LAI of a PacBio-based assembly of common bean[26], indicates a high level of completeness of the repetitive and intergenic regions in our assembly. As has been found in other legumes, LTR-RTs were the predominant TE class in our lablab assembly[15,16,27]. In contrast to findings from lablab's close relatives, however, we found copia LTR-RTs to be more abundant than gypsy LTR-RTs. It is uncommon to see a greater abundance of copia LTR-RTs when compared to gypsy LTR-RTs in plant genomes[28,29], and although the biological significance of elevated copia abundance remains to be seen, further plant genome sequencing will determine whether this finding is indeed a distinguishing feature of lablab.

Lablab has a smaller genome size than other sequenced legumes and also has a smaller number of species-specific orthogroups. Nevertheless, the orthogroup analysis identified several GO categories

enriched in the lablab-specific orthogroups; of particular interest are those involved in fatty acid metabolism, which could underlie seed oil content and composition. In addition, arabinose metabolism genes were enriched in the lablab-unique genes and several other cell wall-related GO terms (specifically related to pectin and lignin) in the orthogroups were expanded in lablab. Cell wall modification could be related to protection from pathogens[30] or drought tolerance[31]. We also show that trypsin inhibitor genes in lablab are arranged in five gene clusters. This defined genome organisation provides a number of opportunities for targeted breeding to reduce trypsin inhibitor content in lablab because of their antinutritional properties. First, screening for copy number variation in these gene clusters, as observed in soybean[32], can help to identify accessions with fewer trypsin inhibitor genes. Secondly, strategies like mutation breeding or genome editing can be used to selectively delete some of these gene

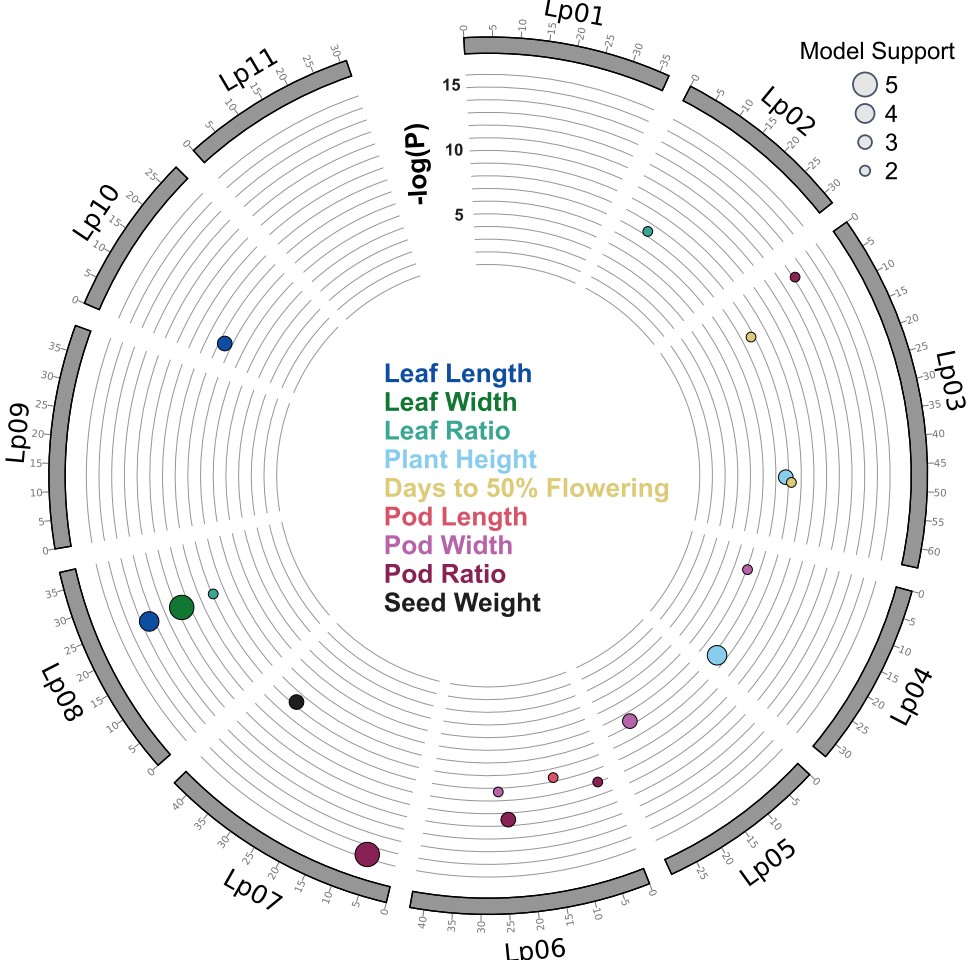

**Fig. 6 | GWAS in lablab.** Circos plots showing the distribution of significant marker-trait associations (MTA) identified in the lablab genome. Only significant markers with adjusted *p*-value > 0.05 (after correcting for multiple comparison using False Discovery Rate approach) from at least two out of five association models tested are shown (Supplementary Table 11). Vertical axis represents −log(raw *p* value). The size of the bubble represents the number of models where the MTA was significant. The colour of the bubble represents the MTA trait. Source data are provided as a Source Data file.

clusters to reduce their antinutritional effects while still maintaining their plant defence function.

A dual origin of domesticated lablab was confirmed, with the localised (to Ethiopia) two-seeded and the widespread four-seeded types being genetically distinct and domestication events occurring in both of these groups. This, therefore, adds lablab to the relatively exclusive list of crops with more than one origin, which includes common bean[15], lychee[33], Tartary buckwheat[34] and, potentially, rice[35] and barley[36]. Data on reproductive isolation between the gene pools are unclear, and crosses are only known between four-seeded samples[37–39], thus any taxonomic reassessment (first suggested by ref. 20) should begin with evaluating reproductive compatibility between the gene pools.

Importantly, our project provides an example for increasing the representation of local researchers in the sequencing of their indigenous crops. Recent studies and commentaries have highlighted the disconnect between the natural distributions of species selected for genome sequencing and the location of the institutions leading their sequencing[10,12,13]. This is particularly true for Africa, where, to date, none of the sequenced indigenous crops was sequenced nor assembled within Africa[12]. Our project breaks this trend because sequencing, assembly and some of the genome analyses were done in Africa, while still recruiting international partners where complementary expertise was beneficial to the project. Thus, we encourage the contribution of

the international community in African orphan crop genomics while supporting more active involvement from local researchers.

Three main features characterised our inclusive genome collaboration - access to low-cost portable sequencing, in-depth capacity building and equitable international collaboration. The high acquisition and maintenance costs of genome sequencing technologies has historically limited the participation of researchers working in LMIC in genome collaborations. Low-cost and portable sequencing platforms such as the ONT MinION, are now making long-read sequencing accessible to researchers in LMIC, thus democratising genome sequencing. Secondly, our project benefited from efforts to build in-depth bioinformatics skills in Africa[13]. Four of the African authors in our study, including two of the first authors, benefited from a residential 8-month bioinformatics training in Africa through the Bioinformatics Community of Practice training (https://acaciaafrica.org). We posit that such in-country and long-term training, as opposed to short workshops, is more effective in developing the high-competence bioinformatics skills that the continent needs. We also acknowledge the efforts by the African Plant Breeding Academy[40] and the recently launched African Biogenome Initiative[12] to develop genomic capabilities on the continent. Lastly, establishing international collaboration helped us to take advantage of existing expertise and already-developed pipelines for genome analyses. With over 20 years of plant genome sequencing, there is now a rich suite of tools, pipelines and

protocols for plant genome analyses. Therefore, African researchers do not have to reinvent the wheel for orphan crop genomics, but instead can form strategic collaborations to access needed expertise, protocols, pipelines and networks.

Our lablab genome assembly and collaboration provide a roadmap for improving agronomic, yield and nutritional traits in other African orphan crops. Given the Africa-centred and inclusive nature of our work, this could be used as a guide by individual labs and multi-national genome consortia including the African Biogenome Initiative[12] to generate high-quality genomic resources for many indigenous species across the continent.

## Methods

### Reference cultivar DNA extraction and sequencing

*Lablab purpureus* (L.) Sweet cv. Highworth[41] seeds were germinated in a petri dish on filter papers moistened with tap water. The sprouted seedlings were transferred to soil and allowed to grow for one month in the greenhouse facility at the International Livestock Research Institute (ILRI, Kenya). Two grams of young trifoliate leaves were harvested, flash-frozen in liquid nitrogen and stored at −80 °C. The leaves were ground in liquid nitrogen using a pestle and mortar and high molecular weight (HMW) DNA was extracted with Carlson lysis buffer (100 mM Tris-HCl, pH 9.5, 2% CTAB, 1.4 M NaCl, 1% PEG 8000, 20 mM EDTA) followed by purification using the Qiagen Genomic-tip 100/G (Qiagen, Cat. No: 10243) based on ONT HMW plant DNA extraction protocol. The library was prepared following the ONT SQK-LSK109 ligation sequencing kit protocol. A total of 1 μg of genomic DNA was repaired and 3′-adenylated with the NEBNext FFPE DNA Repair Mix (NEB, Cat No: M6630) and the NEBNext® Ultra™ II End Repair/dA-Tailing Module (NEB, Cat No: E7595) and sequencing adaptors ligated using the NEBNext Quick Ligation Module (NEB, Cat No: E6056). After library purification with AMPure XP beads (Beckman Coulter, Cat No:A63880), sequencing was conducted at ILRI using the R9.4.1 flow cells on a MinION sequencer platform.

### Genome assembly

Guppy basecaller (ver 4.1.1)[42] was used for base calling the reads using the high-accuracy base-calling model and the resulting fastq files were used for genome assembly. Flye de novo long read assembler[43] (ver 2.7.1) was used for the assembly with the default parameters. The draft assembly was polished with lablab Illumina short reads[11] (NCBI Bioproject PRJNA474418) using HyPo hybrid polisher[44] and scaffolded using Hi-C (see below). The genome assembly quality and completeness were evaluated using QUAST[45] (ver 5.0.2) and BUSCO (ver 5.2.2)[46] where the embryophta_odb10 and fabales lineages were used as references.

### Hi-C scaffolding

Chromatin conformation capture data was generated by Phase Genomics (Seattle, USA) using the Proximo Hi-C 2.0 Kit, which is a commercially available version of the Hi-C protocol. Following the manufacturer's instructions, fresh leaves from lablab were frozen in liquid Nitrogen, ground, crosslinked using a formaldehyde solution and sent to Phase Genomics for library preparation following the manufacturer's protocol. Sequencing of the Hi-C library was performed on an Illumina HiSeq, generating a total of 232,382,372 PE150 read pairs. Reads were aligned to the draft assembly using BWA-MEM (ver 0.7.17-r1198-dirty)[47] with the −5SP and -t 8 options specified, and all other options default. SAMBLASTER (ver 0.1.24)[48] was used to flag PCR duplicates, which were later excluded from the analysis. Alignments were then filtered with SAMtools[50] using the -F 2304 filtering flag to remove non-primary and secondary alignments. Putative misjoined contigs were broken using Juicebox (ver 1.11.08)[51] based on the Hi-C alignments. A total of 6 breaks in 6 contigs were introduced. The same alignment procedure was repeated from the beginning on the resulting corrected assembly. Phase Genomics Proximo Hi-C genome scaffolding platform was used to create chromosome-scale scaffolds from the corrected assembly as described in ref. 52. As in the LACHESIS method[53], this process computes a contact frequency matrix from the aligned Hi-C read pairs, normalised by the number of DPNII restriction sites (GATC) on each contig, and constructs scaffolds in such a way as to optimise expected contact frequency and other statistical patterns in Hi-C data. Approximately 20,000 separate Proximo runs were performed to optimise the number of scaffolds and scaffold construction in order to make the scaffolds as concordant with the observed Hi-C data as possible.

### Synteny-guided chromosome naming

We adopted a naming scheme for the Hi-C-scaffolded chromosomes based on synteny with closely related legumes - *P. vulgaris* (common bean[15]) and *V. unguiculata* (cowpea[16]). For this, we downloaded protein sequence and GFF files of PacBio-based assemblies of *P. vulgaris* (v2.1) and *V. unguiculata* (v1.2) from Phytozome[26] and compared this separately to lablab proteins using BLASTP[54] (settings: -max_target_seqs 1, -evalue 1e-10, -qcov_hsp_perc 70). MCScanX[55] was used to process the individual BLAST output and to detect inter-species collinear blocks.

### Gene annotation

Two annotation pipelines were complementarily used for gene annotation using different evidence support. Protein sequences from five closely related species (*P. vulgaris, V. angularis, C. cajan*, and *M. truncatula*) as well as *Arabidopsis thaliana* were used as protein homology evidence. RNAseq data from lablab cv. Highworth leaves, stem, sepals, and petals[11] were used in de novo transcript assembly with Trinity[56] (ver 2.8.5) and provided as transcript evidence. The Funannotate pipeline[57] (ver 1.8.7) was initially used for gene prediction. Before annotation, the genome assembly was soft-masked, that is, genomic bases at positions with hints of repeats and transposable elements (described below) were transformed to lowercase letters to inform gene prediction software of potential repetitive elements and assist the accurate prediction of protein-coding genes. Using the soft-masked assembly as a reference, Funannotate generated an initial set of gene models using PASA[58] (ver 2.4.1) with RNA-Seq reads, de novo assembled transcripts and protein homology evidence as input. Next, the gene models and protein homology evidence were used to train Augustus[59] (ver 3.3.3), SNAP[60] (ver 2006-07-28) and Glimmerhmm[61] (ver 3.0.4) ab initio gene predictors and predicted genes passed to Evidence modeller[62] (ver 1.1.1) with various weights for integration. tRNAscan-SE[63] (ver 2.0.9) was used to predict non-overlapping tRNAs. Transcript evidence generated from de novo transcript assembly with Trinity (ver 2.8.5) was used to correct, improve and update the predicted gene models and refine the 5′- and 3′-untranslated regions in the final step with Funannotate (ver 1.8.7).

The plant.annot pipeline (https://github.com/PGSB-HMGU/plant.annot) was also used for the prediction of protein-coding genes and incorporated homology information and transcript evidence as well. In the evidence-based step, RNA-Seq data from cv. Highworth leaf, stem, sepal and petal[11] was used for the genome-guided prediction of gene structures. HISAT2[64] (ver 2.1.0, parameter −dta) was used to map RNA-Seq data to the reference genome and the transcripts assembled with Stringtie[65] (ver 1.2.3, parameters -m 150 -t -f 0.3). For the homology-based step, homologous proteins from the closely related species were mapped to the reference genome using the splice-aware mapper GenomeThreader[66] (ver 1.7.1, parameters: -startcodon -finalstopcodon -species medicago -gcmincoverage 70 -prseedlength 7 -prhdist 4). Transdecoder[67] (ver 3.0.0) was used to predict protein sequences and to identify potential open reading frames. The predicted protein sequences were compared to a protein reference database (UniProt Magnoliophyta, reviewed/Swiss-Prot, downloaded from Uniprot on 2017-02-20) using BLASTP[54] (-max_target_seqs 1 -evalue 1e−05).

Conserved protein family domains for all proteins were identified with hmmscan[68] (ver 3.1b2). Transdecoder-predict was run on the BLAST and hmmscan results and the best translation per transcript was selected. Results from the homology and transcript-based gene prediction approaches were combined and redundant protein sequences were removed.

The results from both the Funannotate and plant.annot pipelines were combined as follows. 'bedtools intersect'[69] was used to find overlapping gene models. A BLASTP-search[54] against a database of protein sequences from related species *(P. vulgaris, V. angularis, C. cajan*, and *M. truncatula)* and *A. thaliana* was performed and the best blast hit based on coverage and e-value was selected in case of overlapping gene models. A combined annotation file in gff3-format was created using 'gt merge'[70] and redundant protein sequences as well as non-coding genes were removed. The functional annotation of transcripts as well as the assignment of Pfam[71]- and InterPro[72]-domains, and GO[73,74] terms, were performed using AHRD (automatic assignment of human readable descriptions, https://github.com/groupschoof/AHRD; ver 3.3.3). AHRD assesses homology information to other known proteins using BLASTP searches against Swiss-Prot, The Arabidopsis Information Resource (TAIR), and TrEMBL. The functional annotations are defined using the homology information and the domain search results from InterProScan and Gene Ontology terms. In order to distinguish transposon-related genes from other genes, the functional annotation was used to tag TE-related genes in the genome annotation file. BUSCO[46](ver 5.2.2) was used to assess the completeness of the genome annotation, with sets of universal single-copy gene orthologs from embryophyta and fabales lineages[46].

## Repeat annotation

Repeat annotations for transposable elements (TE) and tandem repeats were conducted independently. For TE annotation, a lablab TE library was constructed using the extensive de novo TE annotator pipeline (EDTA[75] ver 1.9.7). EDTA incorporates both structure and homology-based detection programmes to annotate the predominant TE classes found in plant genomes. EDTA utilises LTRharvest[76], LTR_FINDER[77], LTR_retriever[49], TIR-Learner[78], HelitronScanner[79], RepeatModeler2[80] and RepeatMasker[81] for the identification of TE sequences. The outputs of each module are then combined and filtered to compile a comprehensive, non-redundant TE library. EDTA's inbuilt whole genome annotation function was then used to produce a non-overlapping TE annotation for lablab using the TE library as input. Further calculation of metrics and data visualisation were carried out in R[82] (ver 4.1.2) using the tidyverse suite[83] of packages.

Tandem repeats were identified with TandemRepeatFinder (ver 4.09)[84] under default parameters and subjected to an overlap removal by prioritising higher scores. Higher-scoring matches were assigned first. Lower-scoring hits at overlapping positions were either shortened or removed. Removal was triggered if the lower-scoring hits were contained to $\geq 90\%$ in the overlap or if less than 50 bp of rest length remained.

## Gene family and expansion analysis

Gene families were identified using a genome-wide phylogenetic comparison of the lablab protein sequences and four other legumes. This comprised *P. vulgaris* (PhaVulg1_0), *V. angularis* (Vigan1.1), *C. cajan* (V1.0), and *M. truncatula (*MtrunA17r5). Orthofinder[85] (ver 2.4) was used to identify orthologs and co-orthologs between these species and to group them into gene families. *Arabidopsis thaliana* (Araport 11) was used as an outgroup. The longest transcript was selected for genes with multiple splice variants.

In order to analyse gene family expansion and contraction in lablab, the gene family file produced by Orthofinder was further analysed with CAFE5[86]. An ultrametric tree was built with Orthofinder ($r = 160$) and CAFE5[86] was run with -k 3. Enrichment analysis

using a fisher's exact test ($p_{adj} \leq 0.05$) of significantly ($p$-value of gene family sizes[87] $\leq 0.05$) expanded gene families was performed with TopGO[88].

## Characterisation of the trypsin inhibitor gene family

A set of trypsin inhibitor protein sequences from *Arabidopsis* and four legumes (*Medicago tuncatula, Cajanus cajan, Phaseolus vulgaris*, and *Vigna angularis*) was retrieved from NCBI (XP_020218929.1, XP_027922998.1, CAQ64594.1, NP_001241106.1, NP_001327060.1, AGV54620.1, XP_020224523.1, XP_020224522.1, CAQ64593.1) and used as queries in a blastp search[54] (-max_hsps 1 -evalue 1e-2) against the lablab protein sequences (using the longest transcripts). Protein domain structure in the lablab protein sequences returned from the blast search was searched using InterProScan5[89] (-iprlookup -goterms -pa -f TSV -dp -appl Pfam,TIGRFAM,SUPERFAMIL) and only lablab proteins containing the Bowman–Birk protease inhibitor (PF00228) and Kunitz trypsin inhibitor (PF00197) were retained. In addition, lablab proteins that were not identified from the blastp search but that contain the trypsin inhibitor protein domain were added to the candidate set of lablab trypsin inhibitors. The resulting set of lablab candidate genes was compared to the Orthofinder result and potentially missed orthologs were included. In an additional iteration step, the set of lablab candidate genes was used to search for candidates in the lablab protein set again. Moreover, this set was mapped to the lablab genome with tblastn[54] to search for full-length genes that were not annotated. This however revealed no hits. After identifying the trypsin inhibitor genes in the lablab genome, the gene clusters containing them were extracted from the Orthofinder-results. A multiple sequence alignment of all sequences was calculated using MUSCLE[90] (ver v3.8.1551). A phylogenetic tree was calculated using FastTree[91] (ver 2.1.11) and visualised with iTol[92] (v 6.3). RIdeogram[93] was used to visualise the genomic distribution of lablab trypsin inhibitor genes.

Synteny analyses and the detection of tandemly duplicated genes in the genomes of *Lablab purpureus, Cajanus cajan, Phaseolus vulgaris,* and *Medicago truncatula* was conducted using McScanX[55]. Visualisations were performed with iTOL[92] and circa (http://omgenomics.com/circa).

## Gene expression analysis

In order to analyse gene expression of the trypsin inhibitor genes the SRA-datasets SRR7267957, SRR7267958, SRR7267959, and SRR7267964, corresponding to lablab petal, stem, sepal, and young leaf tissues respectively, were used. Reads were trimmed with fastp[94] (ver 0.23.2). Kallisto[95] (ver 0.44.0) was used to build a transcriptome index. Transcript abundances were quantified using Kallisto (quant with default options). The transcript-level estimates from Kallisto were summarised to the gene level using the tximport package[96] (ver 1.12.3). DESeq2[97] (ver 1.24.0) was used for normalisation and variance stabilising transformation. Pheatmap[98] (ver 1.0.12) was used to visualise gene expression of the trypsin inhibitor gene family in lablab.

## Resequencing and phylogenetic analyses

Seed from lablab plus an outgroup (*Dipogon lignosus*) were obtained from ILRI, the USDA and the Australian Pastures Genebank for the resequencing (Supplementary Data 4). DNA was extracted from leaf tissue using a CTAB-based protocol[99] with minor modifications. In total, 21 accessions of two and four-seeded wild and domesticated lablab and one outgroup were sequenced using $2 \times 150$ bp PE sequencing on an Illumina platform at Novogene (Cambridge, UK). In addition, short read data from lablab cv. Highworth[11], was downloaded from the NCBI Sequence Read Archive (Supplementary Data 4). The reads from all samples were trimmed using Trimmomatic[100] (ver 0.32) with the parameters: ILLUMINACLIP:TruSeq3-PE-2.fa:2:30:10, LEADING:5, TRAILING:5, SLIDINGWINDOW:4:15, MINLEN:72. Between 19.1 and 73.4 M reads remained after trimming. The trimmed reads were

mapped to the chromosome-scale lablab assembly (excluding unmapped contigs) using Bowtie2[101] (ver 2.2.3) and '–very-sensitive-local' settings. SAMtools[50] (ver 1.1) was used to convert.sam to.bam files which were then sorted, and duplicated reads were removed using the Picard toolkit[102] (ver 2.8.3, VALIDATION_STRINGENCY = LENIENT). Depth and coverage were estimated using SAMtools[50] (Supplementary Data 4). Using mpileup from bcftools[103] (ver 1.6.0), the individual sorted bam files were combined into a multi-sample VCF using the settings -Q 13 and -q 10 and variant detection was performed with "bcftools call". After examining the effect of various filtering parameters, variants were subsequently filtered using "bcftools filter", -i'QUAL>20 and DP > 8' expression. Finally, vcftools was used to trim the filtered VCF, removing variants that were missing in more than two samples and those with a minor allele frequency of <5% based on the recommendation of ref. 104. For the phylogenetic analysis, the VCF was filtered to only include variants that were at least 2 kb apart. VCF2Dis (https://github.com/BGI-shenzhen/VCF2Dis/; ver 1.36) was used to create a distance matrix which was submitted to the FAST-ME server (http://www.atgc-montpellier.fr/fastme/) to generate an NJ tree. A total of 1000 replicate matrices were generated in VCF2Dis and the phylip commands 'neighbor' and 'consense' were used to calculate bootstrap values. A second phylogenetic analysis using only variants found in genes with orthologues across the examined legumes gave the same topology (see Results). Genetic diversity for the two subpopulations and $F_{ST}$ between the subpopulations were calculated from the final VCF file (before trimming to remove variants within 2 kb) using vcftools in 100 kb windows. Population genomic analysis was carried out on variants identified as above but excluding the outgroup. The number of populations ($K$) was examined from $K = 1$ to 6 using STRUCTURE[22]. Each $K$ was tested five times with 25,000 permutations after discarding 10,000 as burn-in. The optimal $K$ was then suggested using the Evanno method[105] and StructureHarvester[106].

## Population structure, diversity and association

A total of 2300 seedlings from 203 lablab accessions, that have been maintained at the ILRI forage genebank were grown from seed under screen house conditions at ILRI, Ethiopia. Genomic DNA was extracted from leaves using a DNeasy® Plant Mini Kit (Qiagen, Cat No: 69106). The DNA samples were subjected to genotyping-by-sequencing (GBS) using the DArTseq genotyping platform at Diversity Arrays Technology, Canberra, Australia[107]. We obtained SNP and SilicoDArT marker files from the DArTseq genotyping. The SNP data contain single nucleotide polymorphisms identified between accessions and anchored to the lablab genome. SilicoDArT markers represent mainly presence-absence markers mostly due to structural variations or SNPs at the recognition site of the restriction enzyme used for making the GBS library. The SNP and SilicoDArT markers were filtered based on the marker's minor allele frequency (MAF ≥ 2%), missing values (NA ≤ 10%), independence from each other (linkage disequilibrium/LD ≤ 0.7) and their distribution across the genome. A 2% MAF threshold was used in order to accommodate the smallest sub-population (i.e., two-seeded and four-seeded wild accessions), which make up ~2% of the collection used in the study. This allowed sub-population-specific SNPs to be retained while reducing the chances of retaining SNPs arising from sequencing errors. The 10% missing values were imputed using the missForest R package, running the imputation separately for the major sub-populations: two-seeded, four-seeded wild, and four-seeded cultivated groups.

A pairwise identity-by-descent analysis was conducted using PLINK[108] and contaminants were excluded from the following analyses (see Supplementary Note 1, Supplementary Fig. 12). Genetic diversity was estimated using pairwise Nei's genetic distance[23]. Population stratification was assessed using the Bayesian algorithm implemented in STRUCTURE[22] (ver 2.3.4), in which the burn-in time and the number of iterations were both set to 100,000 with 5 repetitions, testing the

likelihood of 1–20 subpopulations in an admixture model with correlated allele frequencies. Using Structure Harvester[106] the most likely number of subpopulations was determined by the Evanno ΔK method[105]. Accessions with less than 60% membership probability were considered admixed. Hierarchical clustering, principal component analysis (PCA), fixation index ($F_{ST}$) and AMOVA were conducted using the R-packages Poppr[109], adegenet[110] and APE[111].

To examine phenotypic variation among the identified genetic clusters, we used historical phenotype data summarised by Pengelly and Maass[24] and Wiedow[112], in which morpho-agronomic traits on lablab accessions were evaluated in field trials at Ziway site in Ethiopia, in 1998 and 2000, respectively (Supplementary Fig. 13). Quantitative data from the two trials showed high correlation for most traits and were combined using a linear mixed model with the lme4 R package using accession as fixed variable and year of trial as a random variable. Before modelling, the Shapiro-Wilk test in R statistical software (R Core Team, 2021) was used to determine whether the data for each trait had a normal distribution, and a rank-based inverse normal transformation was applied whenever a trait's distribution significantly ($P < 0.01$) departed from normality. Analysis of variance (ANOVA) and Tukey's multiple comparison test were employed to compare significant ($P < 0.05$) phenotypic variation of agro-morphological quantitative traits among the genetic clusters identified by population structure analysis. For qualitative data, a chi-square test was used for similar comparisons among the genetic clusters. The back-transformed data was used for plotting transformed quantitative data in biologically relevant scales.

Marker-trait association analysis was conducted using five statistical models implemented in Genomic Association and Prediction Integrated Tool version 3 (GAPIT3)[113]. The models used include general linear model, mixed linear model (MLM or Q + K), multiple loci mixed model, FarmCPU and BLINK. A total of 21,982 markers (7780 SNPs and 14,202 silicoDArTs), retained after filtering by different criteria as described earlier, were used in the marker-trait association analysis. Markers with false discovery rate adjusted $p$ values < 0.05 were chosen as significant. Only significant markers identified from at least two out of five association models tested are reported..

### Reporting summary

Further information on research design is available in the Nature Portfolio Reporting Summary linked to this article.

## Data availability

Nanopore long reads used for the reference assembly are available from NCBI SRA under BioProject PRJNA824307. Illumina reads for the resequencing samples are available from the NCBI SRA under project number PRJNA834808. The lablab genome and annotation files are available at the Plant Genomics and Phenomics Research Data Repository (e!DAL) [https://doi.org/10.5447/ipk/2022/26][114] and ILRI (International Livestock Research Institute) [https://hpc.ilri.cgiar.org/~bngina/lablab_longread_sequencing_March_2022/]. All lablab accessions are available by request [https://genebank.ilri.org/gringlobal/] Source data are provided with this paper.

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

## Acknowledgements

This research was conducted as part of the CGIAR Research Programme on Livestock, supported by CGIAR Fund Donors. O.S. was supported by the Royal Society FLAIR award (FLR_R1_191850), N.K., H.G. and M.S. were supported by the German Federal Ministry of Education and Research (De.NBI, FKZ 031A536). D.F. was supported by the SoCoBio DTP (grant number BB/T008768/1; BBSRC, UK) to carry out a PhD rotation project in the lab of MAC. D.F. and M.A.C. acknowledge the use of the IRIDIS High Performance Computing Facility, and associated support services at the University of Southampton. I.N., B.W., M.M., D.K. were supported by a Bioinformatics Community of Practice training grant from UK Research and Innovation (UKRI, BB/R020272/1). We also thank Prof Cristobal Uauy for proofreading the manuscript.

## Author contributions

O.S., P.M.F.E., J.D.E., M.A.C., M.S. and C.S.J. conceived and planned the experiments. C.M., L.M. and O.S. performed DNA extraction and Nanopore Sequencing. I.N., B.W., M.M., D.K. and O.S performed the genome assembly. N.K., B.W., M.S., O.S., and I.N. performed genome annotation. D.F. and H.G. annotated the transposable elements and tandem repeats. N.K., B.W. and O.S performed gene family analyses, M.A.C. analysed the re-sequencing data. M.S.M., C.S.J., B.L.M., M.A.C., P.M.F.E and O.S. performed diversity and GWAS analyses on the comprehensive lablab collection. I.N., B.W., N.K., M.S.M., D.F., B.L.M, M.A.C., O.S. and C.S.J. wrote the manuscript. All authors reviewed and approved the final manuscript.

## Competing interests

The authors declare no competing interests.
