## [Peer Review File · Nature Communications]

Chromosome-level genome assembly and population genomic resource to accelerate orphan crop lablab breedingReviewers' Comments:

Reviewer #1:

Remarks to the Author:

Njaci, Waweru, and Kamal et al. report a chromosome scale assembly of the underutilized legume lablab and conducted comparative genomics and population genetic analyses related to domestication and agronomic traits. The authors provide evidence for two domestication events in lablab and provide thousands of loci that can be used for marker assisted breeding. I read this paper with interest, and am happy to see a plant genomics project with most of the effort coming from in-country expertise and long-term training efforts. I have a few relatively minor comments/suggestions that I would like to see addressed.

Line 80. Was the public Illumina data that was used for polishing from the same genotype that was sequenced using ONT data? It is unclear from the methods or results, but this could be problematic if different accessions were sequenced with each technology, as errors could be introduced during polishing due to allelic and structural variation between accessions.

Confirming the two domestication events of lablab is interesting, but I'm not sure why the two seeded accessions that were resequenced by GBS were removed from the analysis. These cultivars should provide additional evidence of the two-domestication event scenario. The authors claim they were removed because of significant missing data, but wouldn't this also have been a major issue with the WGS resequencing data? It would be interesting to learn more about the population divergence of these accessions and links to their geographic origin.

Related to the above comment, do the authors have an estimate of how much global lablab genetic diversity was captured in their study? They found most accessions were collected by the Grassland Research Station in Kenya, but it would be interesting to speculate on how much genetic diversity has not been genotyped yet and if this diversity has agronomic potential.

Line 151. For the nucleotide diversity estimates, were all sites used (including invariant sites) or just the variant sites? It is unclear what percentage of the genome

Minor:

Line 95. Do these additional transcripts represent isoforms or something else?

Versions are provided for most but not all bioinformatics programs (e.g., HyPo hybrid polisher).

Reviewer #2:

Remarks to the Author:

The manuscript by Issaac et al. described sequencing, assembly and resequencing of an African orphan crop, lablab. The genome assembly was much better than the previous published genome sequencing, because of using long reads and HiC sequencing. Based on this improved genome assembly, the authors comprehensively annotated the repeats and protein coding genes, as well as clustered the protein coding genes to gene families. In addition to the assembly and analysis of the reference genome, they also carried out whole genome resequencing on a dozen of individuals to identify genome wide variations and confirm the two independent domestication events based on the constructed phylogenetic tree. Finally, they also applied the DArTseq to identify and genotype representative genetic markers in 138 accessions, as well as illustrate the population stratification according to these markers. Overall, the genome assembly was of high quality, which should benefit future studies on lablab. But the current manuscript failed to clearly present novel findings based on the data generated here, but only to validate previous known findings. I would suggest the authors to focus on the novel findings based on the new genome assembly and the genetic variations identified. I

have two major concerns, as well as some minor suggestions, as listed below.

Major concerns:

1. I don't think it adds much to the research findings by emphasizing the local data production. I admit that it would be better to generate the data locally, especially to help setting up local sequencing facilities and improve the local abilities. However, you might not mention this in long length in a research paper. First of all, there should have been previous sequencing facilities already set up and generating data locally in Africa. For example, a previous news in Nature Biotechnology reported on the 'African coronavirus surveillance network', mentioning about sequencing platforms set up to sequence infectious viruses locally. Secondly, you sequenced Nanopore long reads locally, but you sent samples outside for the resequencing and genotyping. Was the genome assembly carried out locally? This further indicates the unnecessary of emphasizing the local data generation. I strongly disagree that you have developed 'a radically inclusive approach'.

2. One possible reason why they emphasize the local data production, might be that there might not be many novel findings. Even through the genome assembly was quite good, and better than the previous assembly, the authors did not identify novel features in the genome. Results in the assembly part were just descriptive.

Then, in the resequencing part, the authors just confirmed the findings in the previous study that there should be at least two domestication events from two seeded and four seeded progenitors.

There were also descriptive 'results' on the variations in this part.

In the final part of genetic diversity through genotyping, the population stratification should be novel. But this population stratification was only based on limited genetic markers.

3. It was difficult for me to understand the marker part. I am not very familiar with the method used here. It might be better if it can be more clearly described. For example, 41,718 genome wide SNPs were identified and then a subset of them were used for the population analysis. But what were the 73,211 SilicoDArT markers, especially considering that only 57% of them can be mapped to the genome? Except this description, these markers were not further described and used. Also, for the GBS markers, why only 91% can be mapped to the genome? It seems that you were not mapping the reads to the genome assembly, which means even without the genome assembly, you can do this analysis. At least, you need to be clearer on the method of this part.

In the meantime, it was also not clear to me how many individuals were genotyped in this part. What do you mean by genotyping 1,860 individuals from 166 lablab accessions? It would be important to know whether you genotyped 1,860 individuals independently or mixed individuals from the same accession and genotype each accession (mixed individuals).

Minor points:

1) Line 80-81, were the public data generated from the same individual? If not, what was the mapping rate of the short reads to the assembled genome? This can provide information on assembled proportion, in addition to the kmer analysis genome size estimation.

2) Line 84, it would be better to mention the proportion of the assembled sequences to be anchored to chromosomes.

3) Line 86, '61-fold improvement' of what? It would be better to indicate specific statistics used for this assessment in addition to just 'continuity'.

4) Line 100-101, how did you compare the gene models? Comparing to the gene set or the genome assembly should be different. And it would be unfair if you compare the gene sets since you should have applied different annotation pipelines.

5) Line 109, was LAI of 19.8 good or bad? Conclusion should be made here. Also, I don't know whether it was informative to show the LAI of chromosomes in the main text figure (Figure 1D).

6) Line 143-145, what were the coverage when mapping the reads back to the reference genome (In Table S8, coverage was indicated but usually this should be depth or depth coverage; for coverage I am suggesting, it means in the mapping results, what proportion of the assembled genome were covered.)? The coverage also provides information on the genome assembly quality.

- 7) Line 145-146, it was not a good way to map the short reads of other species to the reference genome to determine the SNPs. I would suggest using the assembled genomes to identify the synteny and genotypes within the syntenic blocks can be used for the phylogenetic tree construction. Also, the SNP number seems to be huge, even within the lablab population (more than 15 million SNPs in this ~420 Mb genome). After looking into the method part, I would suggest providing the number of SNPs filtered by bcftools (-i'QUAL>20 & DP>6') which should have excluded low confidence SNPs with low sequencing depth.
- 8) Line 148, although I know you have filtered the SNPs using several criteria, it seems to be quite small amount of the final SNPs used for the phylogenetic analysis. You might need to mention how many SNPs filtered in steps, resulting in only 67,259 SNPs from 15 million SNPs. Were the 67,259 SNPs enough to represent the genetic diversity in the whole genome?
- 9) Line 151-153, how about using structure software to determine the population structure, just like what you have done using the markers in the following section?
- 10) Line 157, 180, 428, Fst was not written correctly.
- 11) Line 189, is it possible to carry out GWAS study based on the phenotypical data? If so, it would greatly improve the novelty of this study.
- 12) Line 318, what do you mean by 'soft-masked'?
- 13) Line 321-322, tRNAs were not mentioned in the main text but just in the method part.
- 14) Line 322-323, transcripts were used to correct the gene models, using what software?
- 15) Line 326-330, what were the mapping rates of the RNA sequencing data? This can reflect the assembly and annotation quality.
- 16) Line 313-339, do you mean you applied two independent pipelines to annotate the protein coding genes? And how did you integrate the results from different pipelines? The other related question was what proportion of the predicted protein coding genes were supported by the RNA sequencing data? Again, this can reflect the genome assembly and gene annotation quality.
- 17) Line 666, why not move the supplementary methods to the methods part?

Reviewer #3:

Remarks to the Author:

Njaci et al unlocked the genome of lablab (*Lablab purpureus*) (cv. Highworth) using Oxford Nanopore technology and Hi-C. They obtained the contig N50 of 11 Mb and constructed synteny-guided pseudomolecules. The genome contains 43% of repeats and over 24,000 high confidence genes. This genomic resource might help to accelerate genomics-based breeding and research. However, the paper needs to address the below concerns in order to use this resource effectively.

Major:

1) The authors have constructed the pseudomolecules using synteny information from close relatives. Still, the quality or accuracy of the pseudomolecule can be seen in the Hi-C contact matrix. However, the authors have not included the Hi-C pseudomolecule plot for the visual inspection of each chromosome. It is highly recommended to validate the contig order and orientation via Hi-C or genetic maps. Gene space assessment can be done at the genome level using BUSCO. Overall, the validation of genome assembly is still lacking. Authors are recommended to perform some analysis and describe the quality of the pseudomolecules in the manuscript.

2) The section containing the evidence for two domestications is superficial. First of all, the size of the population ($n=14$ (lablab lines)) is too small. The SNP calling at the inter-species level is inappropriate to address domestication. The phylogeny pattern that the authors observed is also possible in the case of multiple origins but single domestication (Example: *Oryza sativa*). So, appropriate evidence is needed to support the two domestications in this paper. For example, focusing on genes that were domesticated independently in two gene pools.

3) It is unclear the necessity to use multiple genotypes per accession for diversity analysis though they are inbred. Diversity analysis with only accessions would be more intuitive to the readers. It would also be informative if they compare the diversity between wild and domesticated gene pools. But the population size presented here is still too small.

4) In discussion, the content described in line 240 to 281 are inappropriate for this manuscript. It is better to discuss the appropriate content rather than the general concerns.

Minor:

5) Line 428: (ANOVA)

6) In figures, make uniform patterns such as abc or ABC. Because, figure shows A/B/C but in the legend it is a/b/c.

Reviewer #1 (Remarks to the Author):

Njaci, Waweru, and Kamal et al. report a chromosome scale assembly of the underutilized legume lablab and conducted comparative genomics and population genetic analyses related to domestication and agronomic traits. The authors provide evidence for two domestication events in lablab and provide thousands of loci that can be used for marker assisted breeding. I read this paper with interest, and am happy to see a plant genomics project with most of the effort coming from in-country expertise and long-term training efforts. I have a few relatively minor comments/suggestions that I would like to see addressed.

Line 80. Was the public Illumina data that was used for polishing from the same genotype that was sequenced using ONT data? It is unclear from the methods or results, but this could be problematic if different accessions were sequenced with each technology, as errors could be introduced during polishing due to allelic and structural variation between accessions.

We thank the reviewer for this useful comment. We used Illumina data from the same accession that we used to generate ONT data to avoid including assembly errors, as the reviewer mentioned. We apologise for the lack of clarity in the earlier version of the manuscript. We have now amended this sentence to read as follows “The assembly was polished for error correction using ~380x of publicly available Illumina short reads previously generated from the same accession”.

Confirming the two domestication events of lablab is interesting, but I’m not sure why the two seeded accessions that were resequenced by GBS were removed from the analysis. These cultivars should provide additional evidence of the two-domestication event scenario. The authors claim they were removed because of significant missing data, but wouldn’t this also have been a major issue with the WGS resequencing data? It would be interesting to learn more about the population divergence of these accessions and links to their geographic origin.

We thank the reviewer again for this suggestion. We have now included the 2-seeded and 4-seeded wild accessions in the diversity analyses by strictly filtering for markers with <10% missing data and applying an imputation algorithm separately within each accession type (two- seeded wild, four-seeded wild, and four-seeded cultivated groups). Furthermore, we also included more domesticated lines. We now have 191 accessions for the final diversity analyses instead of the 136 accessions previously used. Our revised diversity analysis supports (Figure 5) the two-domestication hypothesis shown in the phylogenomic analysis (Figure 4; note we have extra accessions in this too).

Related to the above comment, do the authors have an estimate of how much global lablab genetic diversity was captured in their study? They found most accessions were collected by the Grassland Research Station in Kenya, but it would be interesting to speculate on how much genetic diversity has not been genotyped yet and if this diversity has agronomic potential.

It is difficult to estimate how much of the global diversity is captured in our study, however, the diversity analysis was conducted on a collection that was established with the aim to capture as much of the available diversity of lablab as possible, using the tools and information that were available at the time. The collection contains accessions acquired from 25 countries in Africa, Asia, America, Australia and Europe. Also, the inclusion of wild accessions from Africa, and especially the two-seeded cultivated accessions from Ethiopia, increases the diversity of the collection. We therefore believe that

while the collection described in this study is comprehensive, relatively diverse and modest in terms of total number of accessions. To remove any confusion that our collection represents the total global diversity, we have now described the collection as comprehensive, rather than global in the revised manuscript.

Line 151. For the nucleotide diversity estimates, were all sites used (including invariant sites) or just the variant sites? It is unclear what percentage of the genome.

The nucleotide diversity estimates are based on variable sites only. We adjusted the text accordingly.

Minor:

Line 95. Do these additional transcripts represent isoforms or something else?

The additional number of transcripts compared to the gene number represents isoforms/splice variants. We apologise for the lack of clarity and we have adjusted the text for improved clarity.

Versions are provided for most but not all bioinformatics programs (e.g., HyPo hybrid polisher).

Thank you for pointing this out. We added software versions for all the programs used.

Reviewer #2 (Remarks to the Author):

The manuscript by Issaac et al. described sequencing, assembly and resequencing of an African orphan crop, lablab. The genome assembly was much better than the previous published genome sequencing, because of using long reads and HiC sequencing. Based on this improved genome assembly, the authors comprehensively annotated the repeats and protein coding genes, as well as clustered the protein coding genes to gene families. In addition to the assembly and analysis of the reference genome, they also carried out whole genome resequencing on a dozen of individuals to identify genome wide variations and confirm the two independent domestication events based on the constructed phylogenetic tree. Finally, they also applied the DArTseq to identify and genotype representative genetic markers in 138 accessions, as well as illustrate the population stratification according to these markers. Overall, the genome assembly was of high quality, which should benefit future studies on lablab

... But **the current manuscript failed to clearly present novel findings** based on the data generated here, but only to validate previous known findings. I would suggest the authors to focus on the novel findings based on the new genome assembly and the genetic variations identified. I have two major concerns, as well as some minor suggestions, as listed below.

Major concerns:

1. I don't think it adds much to the research findings by emphasizing the local data production. I admit that it would be better to generate the data locally, especially to help setting up local sequencing facilities and improve the local abilities. However, you might not mention this in long length in a research paper. First of all, there should have been previous sequencing facilities already set up and generating data locally in Africa. For example, a previous news in Nature Biotechnology reported on the 'African coronavirus surveillance network', mentioning about sequencing platforms

set up to sequence infectious viruses locally. Secondly, you sequenced Nanopore long reads locally, but you sent samples outside for the resequencing and genotyping. Was the genome assembly carried out locally? This further indicates the unnecessary of emphasizing the local data generation. I strongly disagree that you have developed 'a radically inclusive approach'.

We completely agree that it is better to generate the data locally, and we were attempting to highlight this point as one of the key strengths to our approach. In addition to generating the long-read genome data locally, the genome assembly was also done locally, as well as some of the annotation and diversity analyses. We describe this in the "Authors Contribution" section. However, we take the reviewer's point and fully acknowledge that other sequencing facilities are established on the continent, and particularly used for medical genomics. In the revised manuscript we have tempered the language used and reduced the discussion on our inclusive approach

2. One possible reason why they emphasize the local data production, might be that there might not be many novel findings. Even through the genome assembly was quite good, and better than the previous assembly, the authors did not identify novel features in the genome. Results in the assembly part were just descriptive.

We agree with the reviewer that our genome assembly improves on the previous assembly, particularly in terms of contiguity. Using this chromosome-scale assembly, we have added two detailed analyses which clearly demonstrate how the newly sequenced genome is of use to the community. First, we identified and characterised the trypsin inhibitor (TI) gene family in lablab. TIs are anti-nutritional factors that limit lablab end-use but are also important for the plant's defence against pathogens and herbivorous insects. The chromosome-scale assembly allowed us to show that the lablab TIs are organised in five gene clusters in the genome and that some gene clusters have distinct expression profiles. With this insight, we now propose strategies for targeted breeding to reduce TI content in lablab while preserving their defence functions. Secondly, we now include a GWAS analysis identifying genomic regions underlying variation in important agronomic traits. These two new analyses as well as the other analyses previously described in our manuscript, provide valuable insights for lablab breeding, which would not have been possible without our improved assembly.

Then, in the resequencing part, the authors just confirmed the findings in the previous study that there should be at least two domestication events from two seeded and four seeded progenitors. There were also descriptive 'results' on the variations in this part.

Previous studies were based on very small numbers of markers and/or morphological data, both of which can give erroneous results when analysing domesticated populations, for example if introgression and hybridisation are extensive. By analysing whole genome data we confirm the previous hypothesis robustly and are able to uncover more of the genetic diversity in wild vs. cultivated taxa, a task that had not been carried out. In addition, in the revised version we have added more samples to examine (1) the phylogenetic position of the subspecies *Lablab bengalensis*, and (2) the origin of previously identified 'feral' samples from India. Subsp. *bengalensis* (i.e. are they the wild progenitors of the Indian domesticates?) The findings related to these domesticated and feral samples have implications for studying lablab diversification and hybridisation.

In the final part of genetic diversity through genotyping, the population stratification should be novel. But this population stratification was only based on limited genetic markers.

In the revised version, we have increased the number of quality-filtered markers used for these analyses by three-fold from 2460 to 7780 markers that are distributed across the genome (Figure S4).

The population stratification produced by the increased marker set remains largely the same as described in the previous version of the manuscript (except for the new samples added). We are therefore confident that the number and genome distribution of our markers sufficiently captures the diversity in lablab germplasm.

3. It was difficult for me to understand the marker part. I am not very familiar with the method used here. It might be better if it can be more clearly described. For example, 41,718 genome wide SNPs were identified and then a subset of them were used for the population analysis. But what were the 73,211 SilicoDArT markers, especially considering that only 57% of them can be mapped to the genome? Except this description, these markers were not further described and used. Also, for the GBS markers, why only 91% can be mapped to the genome? It seems that you were not mapping the reads to the genome assembly, which means even without the genome assembly, you can do this analysis. At least, you need to be clearer on the method of this part.

We apologise for not clearly describing the SilicoDArT markers in our previous manuscript. We have now clearly described these markers in our Methods and updated the number and mapping rates of the SilicoDArT markers. We identified 36,803 SilicoDArT markers of which 83% mapped to the genome. SilicoDArT markers represent presence-absence markers mostly due to structural variation or polymorphisms at restriction recognition sites. The relatively low mapping rate observed for SilicoDArT markers compared to SNPs is partly due to the well-described limitation of using one reference for mapping SV¹. Also, the inclusion of wild relatives in our genotyping set contributes to the low mapping rate of SilicoDArT markers as we have shown that these wild relatives diverge significantly from the domesticated (including the reference) lablab genomes. We have now included the mapped SilicoDArT markers in our GWAS analysis and this has allowed us to identify markers (including SilicoDArT markers) underlying important phenotypic variation in lablab.

In the meantime, it was also not clear to me how many individuals were genotyped in this part. What do you mean by genotyping 1,860 individuals from 166 lablab accessions? It would be important to know whether you genotyped 1,860 individuals independently or mixed individuals from the same accession and genotype each accession (mixed individuals).

In the revised version we now clearly describe a two step approach to increase the number of markers in the final analysis while focussing on a large number of accessions. First, owing to the historic nature of the collection, we genotyped 2300 individuals across 203 accessions (previously 1860 individuals across 166 accessions) independently and used these data to identify true-to-type individuals within each accession. We describe this true-to-type analysis in detail in the Supplemental Information. We then parsed down to one representative individual per accession for the diversity analysis described in the main manuscript. We did not mix the DNA from multiple individuals for the diversity analysis. In addition, we used the individual genotype data from multiple plants from some accessions to show that within accession diversity in lablab is low, as expected for a self-pollinating crop.

Minor points:

1) Line 80-81, were the public data generated from the same individual? If not, what was the mapping rate of the short reads to the assembled genome? This can provide information on assembled proportion, in addition to the kmer analysis genome size estimation.

As described in our response to Reviewer1, we used Illumina data for the same accessions used for the long-read sequencing. We apologise for the lack of clarity in the earlier version of the manuscript. We have now amended this sentence accordingly.

2) Line 84, it would be better to mention the proportion of the assembled sequences to be anchored to chromosomes.

We thank the reviewer for this useful comment. We have now included this proportion of the assembled sequence and estimated genome size that were anchored.

3) Line 86, '61-fold improvement' of what? It would be better to indicate specific statistics used for this assessment in addition to just 'continuity'.

Thank you for pointing this out. This has been amended to include the assembly specific contiguity and completeness statistics.

4) Line 100-101, how did you compare the gene models? Comparing to the gene set or the genome assembly should be different. And it would be unfair if you compare the gene sets since you should have applied different annotation pipelines.

We apologise for the misleading phrasing of this statement. The gene set we report in our study has a BUSCO score of 97.3% (embryophyta OrthoDB 10) and is hence likely more complete compared to the previously reported gene set² for which a BUSCO score of 79.4%³ was reported using the same BUSCO-lineage. The BUSCO score of the genome assembly is 98.5% indicating a higher completeness compared to the previous assembly with a BUSCO-score of 93.2%². However, we agree with the reviewer that an in depth comparative analysis of gene sets would be necessary to accurately compare gene numbers. Since this is beyond the scope and goal of our manuscript we removed the statement from the manuscript.

5) Line 109, was LAI of 19.8 good or bad? Conclusion should be made here. Also, I don't know whether it was informative to show the LAI of chromosomes in the main text figure (Figure 1D).

According to the classification system by Ou *et al.*⁴ an LAI (LTR Assembly Index)-score between 10 and 20 is considered reference quality and scores greater than 20 are considered gold-standard. The reported *Lablab purpureus* genome assembly has an LAI-score of 19.8 which indicates a very high 'reference' quality' and very high degree of contiguity. An explanation of the score is now given in the results and discussion sections of the revised manuscript. We believe that Figures 1 C and D are important as they each highlight different aspects of the quality and completeness of our genome assembly and annotations. While Figure 1C shows the high quality and level of completeness of the genome and gene space based on genome and protein BUSCO scores, Figure 1D demonstrates the high level of completeness and contiguity of the repeat space based on the LAI-score of LTR repeat elements. We believe these are relevant and informative metrics for the reader to evaluate the quality of the genome, gene and repeat space in our assembly.

6) Line 143-145, what were the coverage when mapping the reads back to the reference genome (In Table S8, coverage was indicated but usually this should be depth or depth coverage; for coverage I am suggesting, it means in the mapping results, what proportion of the assembled genome were covered.)? The coverage also provides information on the genome assembly quality.

We have added in this suggestion, revealing that average coverage was ca. 85% for the lablab samples and the values for each sample in Supplementary Table 8.

7) Line 145-146, it was not a good way to map the short reads of other species to the reference genome to determine the SNPs. I would suggest using the assembled genomes to identify the synteny and genotypes within the syntenic blocks can be used for the phylogenetic tree construction. Also, the SNP number seems to be huge, even within the lablab population (more than 15 million SNPs in this ~420 Mb genome). After looking into the method part, I would suggest providing the number of SNPs filtered by bcftools (-i'QUAL>20 & DP>6') which should have excluded low confidence SNPs with low sequencing depth.

This is a good point and we simplified the phylogenetic analysis, removing all non-Lablab samples and adding in a more appropriate outgroup. All sites with substantial missing data were removed, yet the analysis comprises a total of 9.8M variants. Given that the samples include two-seeded and four-seeded accessions, which we show are genetically distinct (and should be potentially separated taxonomically), then this number is not unreasonably high. The text now clarifies this number of variants and how this was derived.

8) Line 148, although I know you have filtered the SNPs using several criteria, it seems to be quite small amount of the final SNPs used for the phylogenetic analysis. You might need to mention how many SNPs filtered in steps, resulting in only 67,259 SNPs from 15 million SNPs. Were the 67,259 SNPs enough to represent the genetic diversity in the whole genome?

In addition to our response in point 7 above, we have doubled the number of SNPs in the re-analysis to ~158,000 SNPs.

9) Line 151-153, how about using structure software to determine the population structure, just like what you have done using the markers in the following section?

We have added a structure analysis which further highlights the genetic differentiation between two- and four-seeded groups, plus also identifies differentiation between the wild four-seeded and cultivated four-seeded groups. Two feral samples have been added in the new analysis and it appears they are admixed. This is an interesting and important finding that we now describe in the revised manuscript.

10) Line 157, 180, 428, Fst was not written correctly.

We have corrected this formatting error.

11) Line 189, is it possible to carry out GWAS study based on the phenotypical data? If so, it would greatly improve the novelty of this study.

We thank the reviewer for this suggestion. We have now included a GWAS analysis in the manuscript by combining the SNP and SilicoDArT marker data with the phenotype data. We have now included a figure (Figure 6) that summarises the identified marker-trait associations in the lablab genome. As mentioned by the reviewer, this is a novel analysis that will provide insights for lablab breeding.

12) Line 318, what do you mean by 'soft-masked'?

We have now clarified this in the revised manuscript. "Soft masking" a genome means to convert genomic bases to lowercase letters. This was done at genomic positions with hints of repeats and transposable elements to inform gene prediction software of potential repetitive and/or transposable elements in these regions with the aim to assist the accurate prediction of protein coding genes.

13) Line 321-322, tRNAs were not mentioned in the main text but just in the method part.

Indeed we missed to mention the results of tRNA-content in the results section. We detected 542 tRNA-encoding genes using the funannotate pipeline. We added this to the results section accordingly and provided the annotation file. Thank you for pointing this out.

14) Line 322-323, transcripts were used to correct the gene models, using what software?

As part of the final steps of the Funannotate pipeline (ver 1.8.7), the 'update' step utilises previously generated evidence including *de novo* generated transcripts to refine identified gene models from the previous 'predict' step.

15) Line 326-330, what were the mapping rates of the RNA sequencing data? This can reflect the assembly and annotation quality.

Using default STAR⁵ parameters we could map 89.02% of RNASeq reads (five tissues, 77,010,034 read pairs in total) to the reference genome, of which 74.5% were uniquely mapped. 10.98% of the reads could not be mapped mainly due to too short alignments. Using the same RNASeq data we find transcriptional evidence (tpm \geq 0.5) for 73% of all non-TE related genes (and 62.1% of all genes). These numbers are comparable with other high-quality plant genome sequences and gene annotations⁶⁻⁹ and demonstrate the quality of the lablab genome sequence and gene annotation.

16) Line 313-339, do you mean you applied two independent pipelines to annotate the protein coding genes? And how did you integrate the results from different pipelines?

Indeed, two gene prediction pipelines were applied to improve gene prediction accuracy in the *Lablab purpureus* genome assembly. We apologise for not detailing how the results of the two pipelines were integrated. 'bedtools intersect'¹⁰ was used to find overlapping gene models between both pipelines. To decide which gene model to keep in case of overlapping, non-identical gene models, a blastp search against a database of protein sequences from related species (*P. vulgaris*, *V. angularis*, *C. cajan*, and *M. truncatula*) and *A. thaliana* was performed and the best blast hit based on coverage and e-value was selected. A combined annotation file in gff3-format was created using 'gt merge'¹¹. We have adjusted the text accordingly.

The other related question was what proportion of the predicted protein coding genes were supported by the RNA sequencing data? Again, this can reflect the genome assembly and gene annotation quality.

Using Kallisto¹², the percentage of the non-TE related protein coding genes could be supported by RNASeq data and 62.1% of all genes. These numbers are comparable with other high-quality plant genome annotations⁶⁻⁹. Gene predictions with no expression support may result from ab initio/homology supported models only or are not covered by RNAseq data from currently sampled and available lablab tissues and/or conditions. We incorporated this information into the manuscript.

17) Line 666, why not move the supplementary methods to the methods part?

Thank you for the suggestion. Following *Nature Communication* guidelines, we have moved the methods described in the supplemental information to the Method section, up to the 3000 words limit recommended for this section by *Nature Communication*. The Supplemental Information now only contains supplemental notes that provide additional information on the true-to-type analysis and historical phenotype datasets of the comprehensive lablab collection.

Reviewer #3 (Remarks to the Author):

Njaci et al unlocked the genome of lablab (*Lablab purpureus*) (cv. Highworth) using Oxford Nanopore technology and Hi-C. They obtained the contig N50 of 11 Mb and constructed synteny-guided pseudomolecules. The genome contains 43% of repeats and over 24,000 high confidence genes. This genomic resource might help to accelerate genomics-based breeding and research. However, the paper needs to address the below concerns in order to use this resource effectively.

Major:

1) The authors have constructed the pseudomolecules using synteny information from close relatives. Still, the quality or accuracy of the pseudomolecule can be seen in the Hi-C contact matrix. However, the authors have not included the Hi-C pseudomolecule plot for the visual inspection of each chromosome.

It is highly recommended to validate the contig order and orientation via Hi-C or genetic maps. Gene space assessment can be done at the genome level using BUSCO. Overall, the validation of genome assembly is still lacking. Authors are recommended to perform some analysis and describe the quality of the pseudomolecules in the manuscript.

We apologise for the confusion related to the evaluation of our assembly. We did not use synteny to order our contigs in the first place, but native Hi-C data. We followed an established routine applied in many current high-quality genome assemblies (eg barley and wheat). We now provide the Hi-C¹³ contact map as a supplemental figure. We only used synteny to name the Hi-C-ordered pseudomolecules based on comparison with two closely related legumes. In addition, we provide three universally accepted metrics to show the quality of our assembly. First, we report a BUSCO score of 98.5% at the genome-level against the embryophyta lineage. This put our assembly in the top 5% of BUSCO quality score of all plant genomes sequenced (based on Rose et al, 2021 Nature Plants). Secondly, we also show a BUSCO completeness score of 97.3% at the gene-level suggesting a high degree of completeness of the gene-space in our genome. Finally, the LTR assembly Index (LAI) of our assembly is 19.8 which indicates a reference quality assembly of repeat elements⁴. These metrics together with the Hi-C contact map support a very high level of assembly at the genome, gene and repeat spaces in our study.

2) The section containing the evidence for two domestications is superficial. First of all, the size of the population (n=14 (lablab lines)) is too small. The SNP calling at the inter-species level is inappropriate to address domestication. The phylogeny pattern that the authors observed is also possible in the case of multiple origins but single domestication (Example: *Oryza sativa*). So, appropriate evidence is needed to support the two domestications in this paper. For example, focusing on genes that were domesticated independently in two gene pools.

We thank Reviewer #3 for this comment and we have taken a number of steps to address this comment. First, we remove all non-lablab samples except a (new) outgroup. We performed additional whole genome sequencing of eight new accessions and included these in our phylogeny analysis. Our new analysis supports the two domestication scenario. Adding in the STRUCTURE¹⁴ analysis also adds further support.

Furthermore, we have included 2-seeded (wild and domesticated) and 4-seed wild lablab accessions with those of domesticated accessions in our global diversity analyses which now consists of 191 accessions. Multiple clustering analyses of these data further support the two domestication

hypothesis. Given the strong divergence between the two wild groups we do not believe our data can instead suggest a single domestication.

3) It is unclear the necessity to use multiple genotypes per accession for diversity analysis though they are inbred. Diversity analysis with only accessions would be more intuitive to the readers. It would also be informative if they compare the diversity between wild and domesticated gene pools. But the population size presented here is still too small.

We apologise for not making the purpose of using multiple plants per accession clear in our early manuscript. Given that some of the accessions used in this study have been conserved at the ILRI genbank as far back as 1982, we first needed to examine their genetic purity. We thus primarily genotyped multiple genotypes per accession to determine accessions that are true-to-type for the diversity analysis. We describe this intermediate quality filtering step in detail in our supplemental information. We then used the genotype data from multiple plants to confirm the low within accession diversity we expect from lablab as a self-pollinating plant. Our diversity analysis then included one (true-to-type) plant per accession where more than one was available. As mentioned above, we have both the wild and domesticated gene pools in the diversity analysis and have sequenced and genotyped additional accessions to increase the population size. Importantly, we assert that while our collection might be relatively small, it is relatively diverse because of its origins from different African countries (including Ethiopia - the centre of domestication), India (with) and many other countries outside of Africa.

4) In discussion, the content described in line 240 to 281 are inappropriate for this manuscript. It is better to discuss the appropriate content rather than the general concerns.

We thank the reviewer for this point, however we believe that the content referred to is useful for further discussion. As mentioned in the introduction, our work sets an example for a collaboration model that addresses the under-representation of researchers from Africa (and many low and middle income countries) in the genome sequencing efforts of their native crops as highlighted in recent *Nature* papers^{15,16}. Is it therefore important that we describe to the international community the essential features of our collaboration - portable sequencing platforms, capacity building and equitable South-North collaboration. We hope that our work will encourage more Africa researchers to lead or be more actively involved in the genome sequencing of their native orphan crops. We believe that *Nature Communication* is the perfect platform to raise this discussion given the journal's support for subjects focused on "Agriculture" and "Developing World". We therefore are of the opinion that this section will be of interest to *Nature Communication* readers. However, taking the reviewers' comment on board, we have reduced this section and included more discussion on the scientific findings.

Minor:

5) Line 428: (ANOVA)

Thank you. We have corrected this typing error.

6) In figures, make uniform patterns such as abc or ABC. Because, figure shows A/B/C but in the legend it is a/b/c.

Thank you for pointing this out. We have changed all figure labels to lowercase and meet the journal's requirements for figure layout before publication.

References

1. Li, H. *et al.* Graph-based pan-genome reveals structural and sequence variations related to agronomic traits and domestication in cucumber. *Nat. Commun.* **13**, 682 (2022).
2. Chang, Y. *et al.* The draft genomes of five agriculturally important African orphan crops. *Gigascience* **8**, (2019).
3. Alice, M. *et al.* Genomic data of the Hyacinth Bean (*Lablab purpureus*). (2018) doi:10.5524/101056.
4. Ou, S., Chen, J. & Jiang, N. Assessing genome assembly quality using the LTR Assembly Index (LAI). *Nucleic Acids Res.* **46**, e126 (2018).
5. Dobin, A. *et al.* STAR: ultrafast universal RNA-seq aligner. *Bioinformatics* **29**, 15–21 (2013).
6. Ramírez-González, R. H. *et al.* The transcriptional landscape of polyploid wheat. *Science* **361**, (2018).
7. Kamal, N. *et al.* The mosaic oat genome gives insights into a uniquely healthy cereal crop. *Nature* **606**, 113–119 (2022).
8. International Barley Genome Sequencing Consortium *et al.* A physical, genetic and functional sequence assembly of the barley genome. *Nature* **491**, 711–716 (2012).
9. Vlasova, A. *et al.* Genome and transcriptome analysis of the Mesoamerican common bean and the role of gene duplications in establishing tissue and temporal specialization of genes. *Genome Biol.* **17**, 32 (2016).
10. Quinlan, A. R. & Hall, I. M. BEDTools: a flexible suite of utilities for comparing genomic features. *Bioinformatics* **26**, 841–842 (2010).
11. Gremme, G., Steinbiss, S. & Kurtz, S. GenomeTools: a comprehensive software library for efficient processing of structured genome annotations. *IEEE/ACM Trans. Comput. Biol. Bioinform.* **10**, 645–656 (2013).
12. Bray, N. L., Pimentel, H., Melsted, P. & Pachter, L. Near-optimal probabilistic RNA-seq quantification. *Nat. Biotechnol.* **34**, 525–527 (2016).

13. Padmarasu, S., Himmelbach, A., Mascher, M. & Stein, N. In Situ Hi-C for Plants: An Improved Method to Detect Long-Range Chromatin Interactions. in *Plant Long Non-Coding RNAs: Methods and Protocols* (eds. Chekanova, J. A. & Wang, H.-L. V.) vol. 1933 441–472 (Springer New York, 2019).
14. Pritchard, J. K., Stephens, M. & Donnelly, P. Inference of Population Structure Using Multilocus Genotype Data. *Genetics* vol. 155 945–959 Preprint at <https://doi.org/10.1093/genetics/155.2.945> (2000).
15. Marks, R. A., Hotaling, S., Frandsen, P. B. & VanBuren, R. Representation and participation across 20 years of plant genome sequencing. *Nat Plants* **7**, 1571–1578 (2021).
16. Ebenezer, T. E. *et al.* Africa: sequence 100,000 species to safeguard biodiversity. *Nature* **603**, 388–392 (2022).

Reviewers' Comments:

Reviewer #1:

Remarks to the Author:

The authors have addressed my previous concerns.

Reviewer #2:

Remarks to the Author:

The revised manuscript improved from the previous version, and I found the authors to have addressed all the concerns raised by the reviewers. The authors included more analysis on the assembled genome to indicate the good quality, carried out analysis on the trypsin inhibitors in the assembled genome, added few more samples in the population analysis part, as well as conducted GWAS analysis. However, I still have some major concerns as well as some minor suggestions.

Major concerns:

1. Although the authors mentioned to have 'tempered the language used and reduced the discussion on our inclusive approach', I found most of the discussions related to the 'inclusive approach' remained, probably only to delete 'radically'. From my point of view, what you have mentioned as 'inclusive approach', or 'a model' was actually not novel, because using either Nanopore or PacBio sequencing together with HiC sequencing to establish reference genomes has just become routine. Sequencing locally was not a breakthrough, especially considering about the portability of Nanopore sequencing technology. I only agree that involving local scientists more, and improving the local capabilities in sequencing and bioinformatics, are worthwhile. This study was just among the efforts to promote genomics in Africa (for example, https://pba.ucdavis.edu/PBA_in_Africa/AfPBA_Course_for_Plant_Breeders_932/). Moreover, this is a research article, instead of commentary or communication, so it would be better to focus on the scientific findings. The demonstration effects can be mentioned in the discussion, but for a research article, the significances of data resources or findings should be highlighted instead.
2. I agree that the revised manuscript included major findings including trypsin inhibitors and GWAS study. For trypsin inhibitors, I think the current analysis was not enough. Searching online, I found no previous studies on how the trypsin inhibitors evolved. So in addition to identify the copies of trypsin inhibitors in the assembled genome and related genomes, and depicting their expression in different tissues, I would suggest further analyze how they evolved in the lablab genome or in the legume genomes. Further comparisons of these genes in the synteny blocks of legume genomes, and analyzing their possible involvement with repeats, whole genome duplication or other genome evolution events, would shed lights on how they evolved. More importantly, this would reflect the effectiveness of a good genome assembly, without which these analyses would be difficult.
3. In the part of 'Evidence for two domestications of lablab', I strongly agreed with the other reviewer that the previous, and even the current evidences seemed to be insufficient or at least inappropriate. And I think the suggest of the other reviewer to investigate phylogenetic trees of genes was constructive but not took into consideration by the authors. Looking at Table S8, I found *purpureus2* and *uncinatus2* samples have ~74% coverage, while *purpureus4* samples (closely related to the strain sequenced for reference genome assembly) have ~94% coverage and *uncinatus4* samples have ~85% coverage. This seems quite similar to rice, where *O. rufipogon* is more closely related to *O. sativa japonica* and *O. nivara* is more closely related to *O. sativa indica*. I think you can refer to rice studies, if you would like to provide strong evidence for two independent domestication events.

For me, I have the some more comments on this part. First of all, using the other species as the outgroup should be OK, but mapping the short reads of the other species to the genome of this species was not an appropriate method because of possible artificial mapping. Instead, using the syntenic regions between different species should be better. You can extract population variations in regions in good synteny to the other species, thus the genotypes of the other species in these regions

can be used as outgroup genotypes.

Secondly, I still found the methods were not clear, especially for the method of this part (Resequencing and Phylogenetic Analysis). In total there were 23 individual whole genome sequencing data (as shown in Table S8, 21 lablab individuals and one outgroup sequenced in this study, along with one downloaded from the previous study), were all the 23 individuals subjected to the variation calling procedure mentioned in the methods? Considering about the low mapping rate and low coverage of the outgroup species, it would be erroneous if you included it in the variation calling. You should include all the lablab individuals (include the one you downloaded) but to exclude the outgroup, in order to get proper population variations for lablab.

Finally for this part, I found only 157,913 variations to be used for the phylogenetic analysis. Although I know this set was after filtering and including the outgroup (see the above comment for how I think would be more appropriate to include the outgroup), I would suggest using more SNPs (to exclude the outgroup should result in more variations and should be proper) for STRUCTURE (you can randomly sample subsets from population variations as this software would take too long if you use all variations) and the diversity level calculations.

Other points:

1. Line 1, 30, 37, 55, and 64-65, where the authors emphasized that they have set up a model. As mentioned above, I think it would be better to focus on the data resources and your major findings.
2. In the background, previous genome studies or genetic studies on lablab should be summarized for better understanding on the significances/improvements of this study.
3. Line 74-75, the genome size estimated in the previous genome study was 423 Mb, but they assembled just 385 Mb (395 Mb for scaffolds). Although it is obvious that short reads genome assembly was incomplete possibly with repeats missing, the current assembly was larger than the estimated genome size. Further considering the mapping statistics provided in Table S8 showing ~94% coverage even when the mapping rate can be ~98%, I would suggest investigating the uncovered parts of the genome assembly to see whether they might be mis-assemblies. It should be straightforward, but can reflect the genome assembly quality.
4. Line 94, there is one unnecessary space in between the word ('tissues') and the number indicating the reference.
5. Line 94-95, 'A functional description can be assigned' should be 'Functional descriptions can be assigned'. Usually, some proteins might be assigned multiple functional descriptions.
6. Line 100, 'Copia were the most abundant LTR-RT superfamily' should be 'Copia was the ...'
7. I found the other reviewer mentioned the figure caption problem (for example Figure 1A in maintext but 'a' in Figure 1), which was not fixed despite the authors responded to the reviewer comment mentioning to have 'changed all figure labels'. This is just like some other responds, seemed to be well addressing without proper actions.
8. Line 120-122, were the lablab specific gene families also included in the 448 significantly expanded gene families? Also, why not carry out a GO enrichment analysis on genes from the contracted gene families?
9. Line 148-150, I still think the variations identified were more than expected for this ~420 Mb genome. Please just make sure you have identified the accurate variations.
10. Line 157-158, you should provide the STRUCTURE results (multiple K values, and also the figure showing the optimized K value) as the supplementary information. This can also reflect the domestication process.
11. Line 165, ' 7.06×10^{-5} ' should be ' 7.06×10^{-5} '. Also notice the other places with the same problem.
12. Line 166, 191 and other places, 'Fst' should be 'FST'.
13. Line 174, '2300' should be '2,300'.
14. Line 183-185, again, the STRUCTURE results of multiple K values, as well as determination of optimized K value, should be provided as supplementary information. Furthermore, two sub-populations were observed in the above section according to the whole genome sequencing data.

What was the relationship between these two sub-populations and the seven sub-populations mentioned here?

15. Line 201, for Figure S5, I think it should be more informative to provide the distributions of the traits to see whether they might follow Poisson or other distributions.

16. Line 261-283, as mentioned above, I think this part of the discussion was too long.

17. Line 315, do you mean you sent the DNA to Phase Genomics for Hi-C library construction and maybe sequencing? Is it in Africa? If not, it might not be accurate to claim the genome assembly as 'assembled in Africa'. Saying that, I just mean, you should focus on the resources and findings, as mentioned above as the major concern.

18. Line 455, variations with minor allele frequency $<5\%$ were filtered. But why use 5% as the criteria? With ~ 20 individuals, 5% means just one individual or two alleles, right? Again, as mentioned in my previous comments, it would be informative to see for each step, how many variations were filtered and how many were remained. You did not provide that information although you responded to my previous comment.

19. Figure 5a, labels for y-axis and x-axis were not right.

20. Figure 6, the circus plot was not very straightforward to show the GWAS result. I would suggest Manhattan plots.

Reviewer #3:

Remarks to the Author:

The authors have addressed the concerns that I raised in the review process. The manuscript improved well for publication.

Major concerns 1.

Major concerns:

1. Although the authors mentioned to have ‘tempered the language used and reduced the discussion on our inclusive approach’, I found most of the discussions related to the ‘inclusive approach’ remained, probably only to delete ‘radically’. From my point of view, what you have mentioned as ‘inclusive approach’, or ‘a model’ was actually not novel, because using either Nanopore or PacBio sequencing together with HiC sequencing to establish reference genomes has just become routine. Sequencing locally was not a breakthrough, especially considering about the portability of Nanopore sequencing technology. I only agree that involving local scientists more, and improving the local capabilities in sequencing and bioinformatics, are worthwhile. This study was just among the efforts to promote genomics in Africa (for example, https://pba.ucdavis.edu/PBA_in_Africa/AfPBA_Course_for_Plant_Breeders_932/). Moreover, this is a research article, instead of commentary or communication, so it would be better to focus on the scientific findings. The demonstration effects can be mentioned in the discussion, but for a research article, the significances of data resources or findings should be highlighted instead.

We intentionally structured the discussion section of our manuscript to highlight two mutually inclusive themes that we believe would be of great interest to Nature Communications readers. First, we focus on our scientific findings including the quality of our genome resource, its importance for gene characterisation and breeding in lablab, and the domestication and diversity landscape of lablab. Second, we describe important features that could allow researchers, especially in LMIC countries, to replicate similar genome collaborations to sequence indigenous crops. We strongly believe that these two themes are complementary and can be concurrently discussed in a manuscript. The other reviewers are both in agreement that we have provided the required level of detail which will be of interest to potential readers with diverse interests.

We acknowledge that the use of long-read sequencing has become common-place in many plant genomic projects. However, our point of emphasis is not on the sequencing technology but rather that local researchers from low-income countries conceptualised, sequenced, assembled and analysed the genome of an indigenous crop using accessible sequencing platforms. Such Africa-led efforts are unfortunately not commonplace in plant genomics as was recently described by Mark et al (2022, Nature Plant) and the Africa BioGenome Project (2022, Nature). We believe our manuscript aligns with, and will demonstrate, the recent Nature Groups commitment to address inclusion and ethics in global research (Nature 606, 7 (2022)). We also thank the Reviewer for pointing to the very valuable contributions of other groups and consortiums, including the African Plant Breeding Academy and the AfricaBiogenome project, in increasing genomics capabilities in Africa. We have now included a statement to acknowledge these contributions.

Major concerns 2.

2. I agree that the revised manuscript included major findings including trypsin inhibitors and GWAS study. For trypsin inhibitors, I think the current analysis was not enough. Searching online, I found no previous studies on how the trypsin inhibitors evolved. So in addition to identify the copies of trypsin inhibitors in the assembled genome and related genomes, and depicting their expression in different tissues, I would suggest further analyze how they evolved in the lablab genome or in the legume genomes. Further comparisons of these genes in the synteny blocks of legume genomes, and analyzing their possible involvement with repeats, whole genome duplication or other genome evolution events, would shed lights on how they evolved. More importantly, this would reflect the effectiveness of a good genome assembly, without which these analyses would be difficult.

Additional analyses on the evolution of the trypsin inhibitors in *lablab* have been conducted. We find that 23 of the 35 trypsin inhibitor encoding genes (66%) identified in the *lablab* genome are included within tandemly duplicated gene arrays. Based on the classification of duplicate gene origins of McScanX (Wang et al., 2012), the trypsin inhibitor gene family in *lablab* arose through proximal duplication (p -value 0.0001), as well as tandem gene duplications (p -value $4.65e-13$), but not by whole genome duplication (p -value = 1). Moreover, we analysed the evolution of the trypsin inhibitor gene family in *Vigna angularis*, *Phaseolus vulgaris*, *Medicago truncatula*, and *Cajanus cajan*, and compared it to *Lablab purpureus*. We find that the large cluster on Lp04 (chr4) shows synteny with the other legume genomes and that the five-gene cluster on Lp06 (*Lablab* chr6) is unique to *Lablab* and *Vigna angularis*. We conclude that the synteny within the trypsin inhibitor gene family in *Lablab* is partly conserved between the analysed legume genomes, with additional clusters and gene duplications specific to a few legumes (e.g *Lablab* and *Vigna angularis*). We have added these results and the applied methods to the manuscript, edited Figure 3a to include the visualisation of tandemly duplicated genes and syntenic collinear relationships of genes, and added Figure S5.

Major concerns 3.

Comment 1:

“In the part of ‘Evidence for two domestications of *lablab*’, I strongly agreed with the other reviewer that the previous, and even the current evidences seemed to be insufficient or at least inappropriate. And I think the suggest of the other reviewer to investigate phylogenetic trees of genes was constructive but not took into consideration by the authors.”

And

“... using the other species as the outgroup should be OK, but mapping the short reads of the other species to the genome of this species was not an appropriate method because of possible artificial mapping. Instead, using the syntenic regions between different species should be better. You can extract population variations in regions in good synteny to the other species, thus the genotypes of the other species in these regions can be used as outgroup genotypes.”

The goal of the work is to examine the two domestications hypothesis, and we believe our evidence strongly suggests two origins using multiple approaches. To examine whether mapping the short read data from the outgroup to the *lablab* genome had any effect on our findings, we used the approach of identifying orthologues across multiple legumes and then creating a phylogenetic tree from variants found in only those orthologues. We found this made no difference to the tree topology and was equally well-supported (see the Figure on the right). We have therefore added this information to the main text: “A parallel analysis using only variants from genes which had orthologues in *V. angularis*, *M. truncatula*, *C. cajan*, and *P. vulgaris* gave the same topology.”

Comment 2:

“Looking at Table S8, I found purpureus2 and uncinatus2 samples have ~74% coverage, while purpureus4 samples (closely related to the strain sequenced for reference genome assembly) have ~94% coverage and uncinatus4 samples have ~85% coverage.”

These mapping statistics back up the fact that the 2-seeded accessions are genetically divergent from the 4-seeded accessions and the observation that all 2-seeded have lower mapping % than all 4-seeded again highlights that there are two gene pools. We have highlighted this now in the relevant section as follows: “Mapping and coverage was notably lower for the two-seeded accessions than the four-seeded accessions suggesting genomic divergence between these two gene pools.”

“This seems quite similar to rice, where *O. rufipogon* is more closely related to *O. sativa japonica* and *O. nivara* is more closely related to *O. sativa indica*. I think you can refer to rice studies, if you would like to provide strong evidence for two independent domestication events.”

We thank the reviewer for this suggestion. We already mentioned rice and other species when we discussed other species with multiple domestications : “This, therefore, adds lablab to the relatively ‘exclusive’ list of crops with more than one origin, which includes common bean¹⁵, lychee³⁴, Tartary buckwheat³⁵ and, potentially, rice³⁶ and barley³⁷.”

Comment 3:

“... I still found the methods were not clear, especially for the method of this part (Resequencing and Phylogenetic Analysis)... were all the 23 individuals subjected to the variation calling procedure mentioned in the methods?”

Yes, all individuals were mapped in the same way (21 lablab of ours, one of cv. Highworth and one outgroup). To make this clear, we have edited “The reads were trimmed...” to read “The reads from all samples were trimmed...” in this section of the methods.

Comment 4:

“You should include all the lablab individuals (include the one you downloaded) but to exclude the outgroup, in order to get proper population variations for lablab.”

And

“I would suggest using more SNPs (to exclude the outgroup should result in more variations and should be proper) for STRUCTURE (you can randomly sample subsets from population variations as this software would take too long if you use all variations) and the diversity level calculations.”

And

“Finally for this part, I found only 157,913 variations to be used for the phylogenetic analysis. Although I know this set was after filtering and including the outgroup (see the above comment for how I think would be more appropriate to include the outgroup)”

For the population variation analysis, we already removed the outgroup in our current approach: “Population genomic analysis was carried out on variants identified as above but excluding the outgroup.”. The reason behind the substantial drop in SNP numbers (down to 157,913) is because we thinned the data to only include a maximum of one SNP in each 2kb window. With a genome size of 418 MB (after removing non-chromosome contigs), this represents 76% of the genome (157k SNPs/(418 MB/2k)). We believe this distance-based pruning method is a more appropriate approach (than random sampling) for generating variants for the STRUCTURE and phylogenetic analyses

because tightly linked SNPs will be in LD and presumably give correlated patterns of clustering/relatedness. We therefore think our approach is more likely to give a representative genome-wide analysis.

The reviewer correctly pointed out, however, that the diversity analysis should not be thinned in this way, therefore we have rerun this analysis. We repeated the pipelines excluding the thinning step (but still only including sites with max 2 individuals with missing data and a MAF of 5%) and have revised the text accordingly:

“The same analysis carried out independently for the four-seeded and two-seeded gene pools (and excluding the outgroup) identified 10,666,655 and 5,200,923 variants, respectively.”

And

“Genetic diversity (π per 100 kb window based on only variant sites) was significantly greater (two-sided unpaired T-test, $t = 30.43$, $df = 8095$, $P < 0.001$) in the four-seeded group (0.00790 ± 0.00311 [SD]) than the two-seeded group (0.00599 ± 0.00260 [SD]). Divergence between the two- and four-seeded gene pools was high (mean F_{ST} per 100 kb window = 0.438 ± 0.059 [SD]), which could suggest that these gene pools should be taxonomically re-evaluated as separate taxa.”

Minor concerns

1. Line 1, 30, 37, 55, and 64-65, where the authors emphasized that they have set up a model. As mentioned above, I think it would be better to focus on the data resources and your major findings.

Please see our response to Major Concern 1 above. We believe our manuscript provides sufficient focus on the data resource and scientific findings, while also providing an example for researchers in LMIC in sequencing indigenous crops. However, to remove any notion that we are claiming to be the only group conducting inclusive genome collaboration in Africa, we have replaced the word model with “example”.

2. In the background, previous genome studies or genetic studies on lablab should be summarized for better understanding on the significances/improvements of this study.

We have now added a reference to the previous genome studies in lablab in the background.

3. Line 74-75, the genome size estimated in the previous genome study was 423 Mb, but they assembled just 385 Mb (395 Mb for scaffolds). Although it is obvious that short reads genome assembly was incomplete possibly with repeats missing, the current assembly was larger than the estimated genome size. Further considering the mapping statistics provided in Table S8 showing ~94% coverage even when the mapping rate can be ~98%, I would suggest investigating the uncovered parts of the genome assembly to see whether they might be mis-assemblies. It should be straightforward, but can reflect the genome assembly quality.

Previous studies have shown slight variation in the genome size estimation produced by k-mer-based algorithms. For example, when the genome size of Arabidopsis Col-0 (~135 Mbp) was estimated using five different tools, each tool showed a standard deviation of 2 - 5 Mbp¹. We therefore believe the 3.2 Mbp difference between our total genome length (426.6 Mbp) and the predicted genome size (423 Mbp) is reasonable. Regarding the lower coverage than mapping; when using the genome of a single cultivar (unlike a pan-genome from several cultivars) as a reference for mapping reads from different cultivars, it is not uncommon to have variation in the coverage percentage as shown in several resequencing studies in other legumes. For example, the mapping of resequencing data of 302 wild and cultivated soybean accessions to one reference genome resulted in coverage % from ~77 - 97%². We believe the high (94 - 98%) mapping percentages and variable coverage reported for

our resequencing data does not point to genome mis-assemblies but to a well described limitation of one reference genome to cover the pan-genome space of a species, especially here where we have demonstrated two divergent gene pools are present.

4. Line 94, there is one unnecessary space in between the word ('tissues') and the number indicating the reference.

This is now corrected.

5. Line 94-95, 'A functional description can be assigned' should be 'Functional descriptions can be assigned'. Usually, some proteins might be assigned multiple functional descriptions.

We thank the reviewer for this suggestion. We have changed the sentence accordingly.

6. Line 100, 'Copia were the most abundant LTR-RT superfamily' should be 'Copia was the ...'

We have corrected this sentence accordingly.

7. I found the other reviewer mentioned the figure caption problem (for example Figure 1A in maintext but 'a' in Figure 1), which was not fixed despite the authors responded to the reviewer comment mentioning to have 'changed all figure labels'. This is just like some other responds, seemed to be well addressing without proper actions.

The figure labels in the main text have been changed to match the labels of the figures.

8. Line 120-122, were the lablab specific gene families also included in the 448 significantly expanded gene families? Also, why not carry out a GO enrichment analysis on genes from the contracted gene families?

Lablab specific gene clusters were not included in the significantly expanded gene families. We conducted a GO-enrichment analysis of lablab specific gene families separately and found that they are enriched for fatty acid biosynthesis and arabinose metabolism and are involved in pollen-pistil interactions and general plant development (Table S6).

We added a GO-enrichment analysis for contracted gene families to the manuscript. We can find that gene families involved in amide biosynthetic and metabolic processes are contracted. This has been added as Figure S4.

9. Line 148-150, I still think the variations identified were more than expected for this ~420 Mb genome. Please just make sure you have identified the accurate variations.

This is correct and includes the two divergent gene pools and the outgroup (23 samples), but does not exclude singletons or sites with missing data. See addition in major comment 3 for the gene pool specific variants, which is about half of the value for the 23 samples.

10. Line 157-158, you should provide the STRUCTURE results (multiple K values, and also the figure showing the optimized K value) as the supplementary information. This can also reflect the domestication process.

We thank the reviewer for this suggestion. We have added it (Figure S6) and we mention the most obvious additional finding that comes out from this in the text now "Other values of K (the number of clusters) differentiate the likely feral accessions from the other four-seeded accessions (Figure S6)."

11. Line 165, '7.06 x 10-5' should be '7.06 × 10⁻⁵'. Also notice the other places with the same problem.

We have now used the correct decimal form.

12. Line 166, 191 and other places, 'F_{st}' should be 'F_{ST}'.

This has now been corrected.

13. Line 174, '2300' should be '2,300'.

This has now been corrected.

14. Line 183-185, again, the STRUCTURE results of multiple K values, as well as determination of optimized K value, should be provided as supplementary information. Furthermore, two sub-populations were observed in the above section according to the whole genome sequencing data. What was the relationship between these two sub-populations and the seven sub-populations mentioned here?

We thank the reviewer for this suggestion. We have now provided the plots showing the determination of the optimum K as a supplemental Figure (Figure S8). We have also included the STRUCTURE plots for different K clusters in the main Figure 5. The optimum K of 2 obtained from our STRUCTURE analysis fully supports the two sub-populations (2-seeded and 4-seeded) obtained from the phylogenetic tree. The additional K from our STRUCTURE plots further highlight other sub-populations in the 4-seeded group. We have now included a statement describing concordance between the results obtained from our whole genome and reduced representation datasets.

15. Line 201, for Figure S5, I think it should be more informative to provide the distributions of the traits to see whether they might follow Poisson or other distributions.

We have now included supplemental Figure S12 that shows the distribution of the traits examined. We also provided the raw data that was used to create this plot in the source data. As we previously mentioned in our manuscript, quantitative traits that did not follow a normal distribution were transformed prior to statistical and GWAS analyses.

16. Line 261-283, as mentioned above, I think this part of the discussion was too long.

As we mentioned previously, we believe that this section adds to the value of our manuscript and would be of interest to Nature Communication readers without taking away from the technical discussion on the data resource and scientific findings.

17. Line 315, do you mean you sent the DNA to Phase Genomics for Hi-C library construction and maybe sequencing? Is it in Africa? If not, it might not be accurate to claim the genome assembly as 'assembled in Africa'. Saying that, I just mean, you should focus on the resources and findings, as mentioned above as the major concern.

The NGS data used to make our assembly was sequenced and assembled in Africa by local researchers at the International Livestock Research Institute, Kenya. Hence the reason we mentioned that the genome was assembled in Africa. To scaffold this base assembly into pseudomolecules, we initially attempted HiC scaffolding by using a commercial kit manufactured by Phase Genomics (USA) to make, sequence and analyse HiC libraries locally at ILRI Kenya. This attempt was unsuccessful due to technical difficulties with the HiC library preparation. We therefore sent fixed tissue to the kit manufacturer (Phase Genomics) to use their established pipeline and proprietary tools for the Hi-C

scaffolding as a service. As we mentioned in our discussion, we encourage local researchers to use existing expertise and already-developed pipelines to advance their genomics project.

18. Line 455, variations with minor allele frequency $< 5\%$ were filtered. But why use 5% as the criteria? With ~ 20 individuals, 5% means just one individual or two alleles, right? Again, as mentioned in my previous comments, it would be informative to see for each step, how many variations were filtered and how many were remained. You did not provide that information although you responded to my previous comment.

It has indeed been shown that different MAF cutoffs affect results of population-based studies, but that "... model-based inference of population structure is confounded when singletons are included in the alignment, and that both model-based and multivariate analyses infer less distinct clusters when more stringent MAF cutoffs are applied."³ Given this background, we believe the 5% MAF threshold we applied sufficiently removed singletons (as pointed out by the reviewer) without being too stringent to infer less distinct clusters.

19. Figure 5a, labels for y-axis and x-axis were not right.

We have modified the axis label accordingly.

20. Figure 6, the circos plot was not very straightforward to show the GWAS result. I would suggest Manhattan plots.

We identified marker trait associations across nine traits using 2 - 5 different statistical models. Due to space constraints, the high number of plots (18 - 45) required to show these associations would not fit into the main manuscript. We therefore presented a summary of these associations in the main manuscript using a circos plot (Figure 6). However, following the reviewer's suggestion, we have now provided a supplemental figure (Figure S11) showing the Manhattan plot of the GWAS results obtained from one of the five models examined for all the traits.

References

1. Sun, H., Ding, J., Piednoël, M., and Schneeberger, K. (2018). *findGSE: estimating genome size variation within human and Arabidopsis using k-mer frequencies. Bioinformatics 34, 550–557.*
2. Zhou, Z., Jiang, Y., Wang, Z., Gou, Z., Lyu, J., Li, W., Yu, Y., Shu, L., Zhao, Y., Ma, Y., et al. (2015). *Resequencing 302 wild and cultivated accessions identifies genes related to domestication and improvement in soybean. Nat. Biotechnol. 33, 408–414.*
3. Linck, E., and Battey, C.J. (2019). *Minor allele frequency thresholds strongly affect population structure inference with genomic data sets. Mol. Ecol. Resour. 19, 639–647.*

Reviewers' Comments:

Reviewer #2:

Remarks to the Author:

All the previously raised points have been addressed. Thanks.